# Linear ubiquitination induces NEMO phase separation to activate NF-κB signaling

Simran Goel[1] , Rosario Oliva[2,3], Sadasivam Jeganathan[1] , Verian Bader[1,8] , Laura J Krause[1,7] , Simon Kriegler[2], Isabelle D Stender[4], Chadwick W Christine[5], Ken Nakamura[5,6], Jan-Erik Hoffmann[4], Roland Winter[2,7], Jörg Tatzelt[7,8] , Konstanze F Winklhofer[1,7]

The NF-κB essential modulator NEMO is the core regulatory component of the inhibitor of κB kinase complex, which is a critical checkpoint in canonical NF-κB signaling downstream of innate and adaptive immune receptors. In response to various stimuli, such as TNF or IL-1β, NEMO binds to linear or M1-linked ubiquitin chains generated by LUBAC, promoting its oligomerization and subsequent activation of the associated kinases. Here we show that M1-ubiquitin chains induce phase separation of NEMO and the formation of NEMO assemblies in cells after exposure to IL-1β. Phase separation is promoted by both binding of NEMO to linear ubiquitin chains and covalent linkage of M1-ubiquitin to NEMO and is essential but not sufficient for its phase separation. Supporting the functional relevance of NEMO phase separation in signaling, a pathogenic NEMO mutant, which is impaired in both binding and linkage to linear ubiquitin chains, does not undergo phase separation and is defective in mediating IL-1β–induced NF-κB activation.

## Introduction

NEMO, also known as IKKγ, is the core regulatory component of the IκB kinase (IKK) complex, which is a key player in NF-κB pathway activation in response to various stimuli (Israel, 2010; Clark et al, 2013; Maubach et al, 2017). NEMO has no enzymatic activity but is essential for the activation of IKKα and IKKβ in canonical NF-κB signaling. A crucial step in IKK complex activation is the binding of NEMO to M1- and K63-linked polyubiquitin and the covalent attachment of K63- and M1-linked ubiquitin to NEMO (Haas et al, 2009; Rahighi et al, 2009; Tokunaga et al, 2009). This triggers a conformational change and the oligomerization of NEMO, thereby activating the associated kinases IKKα and IKKβ. Phosphorylation of the NF-κB inhibitory protein IκBα by IKKα/β leads to its modification

with K48-linked ubiquitin chains and subsequent proteasomal degradation of IκBα. NF-κB heterodimers, typically p65 and p50, liberated from their inhibitory binding to IκBα, can then translocate from the cytoplasm to the nucleus to regulate expression of NF-κB–dependent genes in a cell type–dependent and context-dependent manner (Hayden & Ghosh, 2014; Brenner et al, 2015; Annibaldi & Meier, 2018; Varfolomeev & Vucic, 2018).

Linear or M1-linked polyubiquitination is characterized by the head-to-tail linkage of ubiquitin molecules via the C-terminal carboxyl group of the donor ubiquitin and the N-terminal methionine of the acceptor ubiquitin (Kirisako et al, 2006). The assembly of linear ubiquitin chains is catalyzed by the linear ubiquitin chain assembly complex (LUBAC), consisting of the catalytic subunit HOIP (HOIL-1-interacting protein or RNF31), HOIL-1L (heme-oxidized IRP2 ubiquitin ligase 1L or RBCK1), and SHARPIN (SHANK-associated RH domain interactor) (Dittmar & Winklhofer, 2019; Spit et al, 2019; Oikawa et al, 2020; Fiil & Gyrd-Hansen, 2021; Fuseya & Iwai, 2021; Jahan et al, 2021; Shibata & Komander, 2022; Tokunaga & Ikeda, 2022). Several ubiquitination sites for M1-linked ubiquitin (K285, K309) and K63-linked ubiquitin (K285, K321, K325, K326, and K399) have been identified in NEMO (Huang et al, 2003; Abbott et al, 2004; Zhou et al, 2004; Tokunaga et al, 2009; Ikeda et al, 2011). Moreover, K63/M1 heterotypic chain formation has been observed in NF-κB activation downstream of several innate immune receptors, which allows the co-recruitment of regulatory factors binding preferentially either to K63-linked ubiquitin, such as the TAK1 (TGFβ-activated kinase) complex, or to M1-linked ubiquitin, such as the IKK complex (Emmerich et al, 2013, 2016; Fiil et al, 2013; Hrdinka et al, 2016; Cohen & Strickson, 2017).

To facilitate the swift and reversible assembly of signaling complexes, higher-order oligomerization of signaling components is an important aspect of the spatiotemporal regulation of cellular signaling (Wu, 2013). An emerging mechanism underlying the organization of signaling components is the formation of biomolecular condensates through liquid-liquid phase separation (LLPS),

[1]Department Molecular Cell Biology, Institute of Biochemistry and Pathobiochemistry, Ruhr University Bochum, Bochum, Germany  [2]Physical Chemistry I-Biophysical Chemistry, Department of Chemistry and Chemical Biology, TU Dortmund University, Dortmund, Germany  [3]Department of Chemical Sciences, University of Naples Federico II, Naples, Italy  [4]Protein Chemistry Facility, Max Planck Institute of Molecular Physiology, Dortmund, Germany  [5]Department of Neurology, UCSF, San Francisco, CA, USA  [6]Gladstone Institute of Neurological Disease, Gladstone Institutes, San Francisco, CA, USA  [7]RESOLV Cluster of Excellence, Ruhr University Bochum, Bochum, Germany  [8]Department Biochemistry of Neurodegenerative Diseases, Institute of Biochemistry and Pathobiochemistry, Ruhr University Bochum, Bochum, Germany

Correspondence: konstanze.winklhofer@ruhr-uni-bochum.de; joerg.tatzelt@rub.de

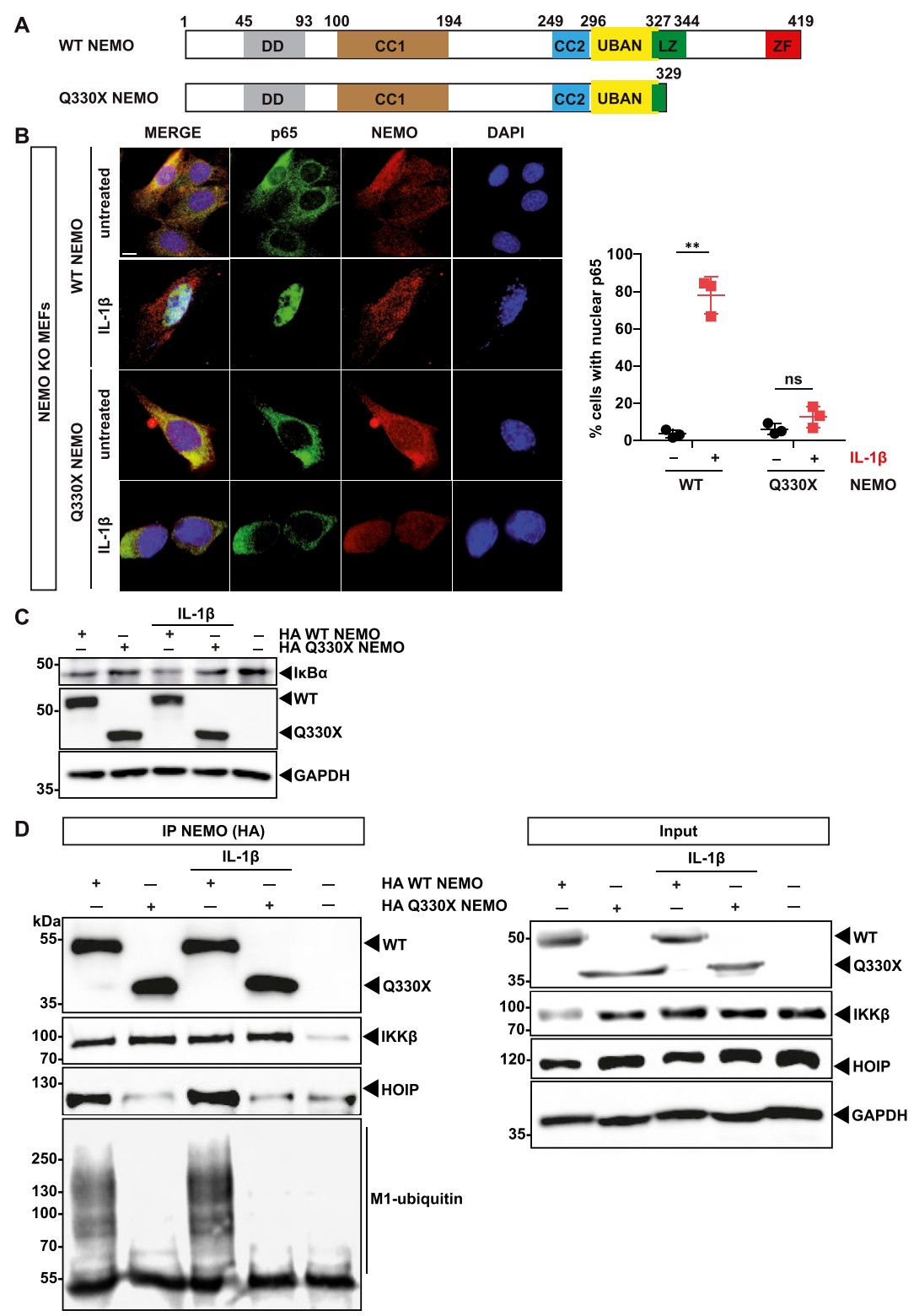

**Figure 1. The pathogenic Q330X NEMO mutant is defective in NF-κB signaling.**
**(A)** Domain structure of WT human NEMO and of the pathogenic mutant Q330X NEMO. DD, dimerization domain (aa 45–93); CC1, coiled-coil 1 domain (aa 100–194); CC2, coiled-coil 2 domain (aa 249–292); UBAN, ubiquitin binding in ABIN and NEMO (aa 296–327); LZ, leucine zipper (aa 322–344); ZF, zinc finger (aa 389–419). **(B)** Q330X NEMO does not promote p65 translocation upon IL-1β stimulation. NEMO KO MEFs were reconstituted with HA-tagged WT NEMO or Q330X NEMO. 24 h later, the cells were stimulated with murine IL-1β (20 ng/ml) for 15 min or left untreated and then analyzed by immunocytochemistry and SR-SIM using antibodies against NEMO and p65 (scale bar, 10 µm). Data are displayed as mean ± SD and were analyzed by Kruskal-Wallis test followed by Dunn's Multiple Comparison Test, n = 3, **P ≤ 0.01. **(C)**. WT NEMO but not Q330X NEMO induces IκBα degradation upon IL-1β stimulation. HEK293T cells were transiently transfected with HA-tagged WT NEMO or HA-Q330X NEMO, as

driven by dynamic interactions between multivalent molecules (Shin & Brangwynne, 2017; Martin & Mittag, 2018; Alberti & Dormann, 2019; Case et al, 2019; Gomes & Shorter, 2019; Choi et al, 2020; Zbinden et al, 2020; Mittag & Pappu, 2022). Thermodynamically, phase separation is governed by the minimization of the global free energy, which involves contributions from enthalpy and entropy. At the same time, inter- and intramolecular protein interactions are maximized, increasing the local concentration of biomolecules and thereby enhancing the efficiency of catalytic reactions. In addition, biomolecular-solvent interactions modulate phase transitions.

It has been shown previously that activation of NF-κB signaling by TNF or IL-1β induces the formation of NEMO-positive punctate structures in a ubiquitin-dependent manner (Tarantino et al, 2014; Scholefield et al, 2016). Heterotypic multivalent interactions with polyubiquitin can induce LLPS of shuttle proteins involved in protein quality control, such as p62 and RAD23B (Danieli & Martens, 2018; Dao & Castaneda, 2020). We therefore addressed the question of whether NEMO can undergo phase separation and whether such a process is influenced by ubiquitin. Here we show that M1-linked ubiquitin chains generated by HOIP induce phase separation of NEMO in vitro, which correlates with the formation of NEMO condensates in cells. NEMO foci formation and efficient NF-κB pathway activation were abrogated in the absence of HOIP or when NEMO mutants defective in binding or linkage to M1-ubiquitin were expressed, suggesting a coupling of NEMO phase separation and downstream signaling.

## Results

### The pathogenic mutant Q330X NEMO is defective in activating NF-κB signaling

We have identified a novel pathogenic mutation in the *IKBKG* gene encoding NEMO in a patient suffering from *Incontinentia pigmenti* (IP). IP is a rare X-linked ectodermal dysplasia, which is usually lethal in males and shows a wide spectrum of clinical manifestations in females (Smahi et al, 2000; Conte et al, 2014; Fusco et al, 2015). The premature stop codon in Q330X NEMO results in the loss of the C-terminal zinc finger (ZF) domain (residues 389–419) and a partial loss of the leucine zipper (residues 329–343) (Fig 1A).

To functionally characterize Q330X NEMO in a cellular model, we treated NEMO KO MEFs reconstituted by either WT NEMO or Q330X NEMO with IL-1β for 15 min. The cells were analyzed for nuclear translocation of the NF-κB subunit p65 by immunocytochemistry and super-resolution structured-illumination microscopy (SR-SIM). Whereas IL-1β treatment induced nuclear translocation of p65 in about 80% of cells expressing WT NEMO, no significant increase in nuclear p65 was seen in cells expressing Q330X NEMO (Fig 1B). Moreover, IκBα degradation upon IL-1β treatment was strongly

impaired in cells expressing Q330X NEMO in contrast to WT NEMO-expressing cells, confirming the inability of mutant NEMO to promote NF-κB pathway activation (Fig 1C).

Engagement of IL-1β with its receptor at the plasma membrane results in the recruitment of LUBAC to the signaling complex, where HOIP modifies its substrates, such as IRAK1/4 and MyD88, with M1-linked ubiquitin chains (Emmerich et al, 2013). Binding of NEMO to M1-linked ubiquitin chains is a crucial event in the activation of the IKK complex. In some conditions, NEMO is also covalently modified with M1-linked ubiquitin chains, although this has not been shown for endogenous NEMO upon IL-1β receptor activation (Emmerich, 2013). NEMO interacts with HOIP upstream of the UBAN (ubiquitin binding in ABIN and NEMO) domain and with M1-linked ubiquitin via its UBAN domain, whereas binding to IKKα/β occurs through the N-terminal region of NEMO (residues 40–120) (Tokunaga et al, 2009; Clark et al, 2013; Fujita et al, 2014; Rahighi et al, 2022). To test whether these interactions are impaired in Q330X NEMO, we performed co-immunoprecipitation experiments. Lysates of cells expressing either WT or Q330X HA-tagged NEMO were affinity-purified via its HA tag and immunoblotted by using IKKβ, HOIP, or M1-ubiquitin-specific antibodies. HOIP and M1-linked ubiquitin chains co-purified only with WT NEMO but not with Q330X NEMO, whereas IKKβ bound to both WT NEMO and Q330X NEMO (Fig 1D). Notably, increased expression of NEMO is sufficient to promote association with linear ubiquitin chains, even in the absence of additional stimuli, such as TNF or IL-1β, explaining the M1-ubiquitin–positive signal in WT NEMO–expressing cells (Fig 1D, lane 1). Taken together, Q330X NEMO is defective in interacting with HOIP and linear ubiquitin chains and does not promote NF-κB pathway activation in response to IL-1β stimulation.

### IL-1β induces the formation of cytosolic NEMO assemblies in an M1-ubiquitin–dependent manner

It was previously described that IL-1β induces the formation of cytoplasmic NEMO assemblies (Tarantino et al, 2014; Scholefield et al, 2016). To test whether Q330X NEMO shows this behavior, NEMO KO MEFs reconstituted by either WT NEMO or Q330X NEMO were treated with IL-1β for 15 min. The cells were permeabilized by saponin buffer before fixation, thereby partially extracting soluble NEMO without affecting IL-1β–induced NEMO assemblies (Tarantino et al, 2014), and then analyzed by immunocytochemistry and SR-SIM. Upon IL-1β stimulation, WT NEMO formed assemblies, which were not observed in cells expressing Q330X NEMO (Fig 2A). We also tested the K285/K309R NEMO mutant, which lacks two lysine residues described to be covalently modified with linear ubiquitin chains (lysine residues at position 285 and 309 replaced by arginines) (Tokunaga et al, 2009). Similarly to Q330X NEMO, the K285/K309R NEMO mutant was defective in forming NEMO foci and IKK

---

indicated. 1 d after transfection, the cells were stimulated with human IL-1β (25 ng/ml) for 15 min or left untreated and analyzed by immunoblotting using antibodies against IκBα, NEMO, and GAPDH. **(D)** The Q330X mutation disrupts binding of NEMO to HOIP and M1-linked ubiquitin. HEK293T cells were transiently transfected with HA-tagged WT NEMO or Q330X NEMO, as indicated. 1 d after transfection, the cells were stimulated with human IL-1β (25 ng/ml) for 15 min or left untreated and then lysed. HA-tagged NEMO was immunoprecipitated using anti-HA-beads, followed by immunoblotting using antibodies against HA, IKKβ, HOIP, and M1-ubiquitin. The input was immunoblotted for NEMO, IKKβ, HOIP, and GAPDH.
Source data are available for this figure.

complex activation upon IL-1β receptor activation (Fig 2A). In addition, increased expression of OTULIN, the only deubiquitinase that exclusively hydrolyzes linear polyubiquitin chains (Keusekotten et al, 2013; Rivkin et al, 2013), interfered with IL-1β–induced nuclear translocation of p65, suggesting that M1-linked ubiquitin chains are required for NF-κB pathway activation in this context (Fig 2B). Moreover, in HOIP CRISPR/Cas9 KO SH-SY5Y cells, IL-1β–induced cytoplasmic foci formation of endogenous NEMO was not observed, in contrast to WT SH-SY5Y cells (Fig 2C, left panel). A corresponding immunoblot analysis indicated that NF-κB pathway activation was compromised upon IL-1β receptor activation in the absence of HOIP, indicated by reduced phosphorylation of IKKs (Fig 2C, right panel). Notably, the abundance of endogenous NEMO was decreased in HOIP KO SH-SY5Y cells, most probably by a reduced stability of NEMO not being recruited to LUBAC. These experiments showed that the formation of NEMO assemblies in IL-1β–treated cells requires formation of M1-linked ubiquitin chains, suggesting that NEMO condensation is a prerequisite for downstream signaling leading to the activation of NF-κB.

## HOIP-induced linear ubiquitination is required for NEMO phase separation

In the next step, we used in vitro systems to study NEMO condensate formation with purified components. To test whether ubiquitination of NEMO can induce its phase separation, we performed an in vitro linear ubiquitination assay by incubating untagged WT NEMO or Q330X NEMO purified from insect cells with the ubiquitin-activating enzyme UBE1, the E2 ubiquitin-conjugating enzyme UB2D3 (UbcH5c), and catalytically active C-terminal HOIP comprising the RBR (RING-in-between-RING) and LDD (linear ubiquitin chain determining domain) domains (Stieglitz et al, 2012). Bright-field microscopy revealed that WT NEMO undergoes phase separation, forming assemblies with a regular spherical shape upon HOIP-mediated linear ubiquitination, which was not observed for Q330X NEMO (Fig 3A). The NEMO assemblies formed by linear ubiquitination were highly mobile and dynamic, evidenced by fusion events captured by bright-field microscopy (Fig 3B).

In parallel, the reaction mixtures were analyzed by SDS–PAGE followed by immunoblotting using NEMO- and M1-ubiquitin–specific antibodies. A smear of higher molecular weight species was detected in the sample containing WT NEMO, indicative of covalent modification of NEMO with M1-linked ubiquitin chains by HOIP (Fig 3C). However, this smear did not occur when Q330X NEMO was analyzed, suggesting that HOIP was not able to ubiquitinate Q330X NEMO (Fig 3C). A corresponding immunoblot using M1-ubiquitin-specific antibodies revealed the ATP-dependent formation of linear ubiquitin chains in both samples, indicating that HOIP effectively assembled free linear polyubiquitin chains (Fig 3C). In conclusion, HOIP promotes phase separation of WT NEMO by ubiquitination and/or by generating unanchored M1-linked ubiquitin chains which bind to NEMO.

## Binding of M1-linked ubiquitin chains promotes phase separation of NEMO

To address the mechanism underlying phase separation of NEMO in more detail, we expressed NEMO-GFP fusion proteins containing an N-terminal MBP (maltose-binding protein) to keep the recombinant proteins soluble (Fig S1A). After purification from *Escherichia coli*, the N-terminal MBP as well as the C-terminal His$_6$ tags were cleaved off by Tobacco Etch Virus (TEV) protease (Kamps et al, 2021) (Fig 4A). Under these conditions, both WT NEMO-GFP and Q330X NEMO-GFP were soluble (Fig 4B). We also performed size exclusion chromatography with untagged WT NEMO and Q330X NEMO and observed similar profiles in the calibrated range of the column, but we cannot exclude possible differences in oligomeric species of higher molecular weight (Fig S1B). Phase separation of WT and mutant NEMO did not occur in different buffers at various pH conditions (Fig S2A and B).

Next, we tested whether binding of M1-linked polyubiquitin to NEMO is sufficient to induce its phase separation. Upon mixing NEMO with recombinant M1-linked tetra-ubiquitin (4×M1-ub), phase separation of both WT NEMO-GFP and untagged NEMO was observed, which was not seen when WT NEMO was mixed with mono-ubiquitin (Figs 4C–E and S2C and D). Under all conditions tested, M1-linked tetra-ubiquitin did not induce phase separation of untagged Q330X NEMO or Q330X NEMO-GFP (Figs 4D and E and S2C and D). The formation of WT NEMO condensates was dependent on the concentrations of both NEMO and M1-linked tetra-ubiquitin, and larger assemblies formed with increasing concentrations (Fig 4C). Interestingly, a similar behavior was observed by Du et al upon incubation of recombinant WT NEMO with K63-linked polyubiquitin chains (Du et al, 2022). Taken together, phase separation of WT NEMO was induced by binding to M1-linked tetra-ubiquitin but not to mono-ubiquitin, suggesting an essential role for multivalent heterotypic interactions. In contrast, Q330X NEMO did not undergo phase separation in the presence of M1-linked ubiquitin chains. In addition, we found salt-induced reentrant phase behavior for both WT NEMO and pathological Q330X NEMO (Fig S2E). Notably, the reentrant phase transition we report here takes place at salt concentrations higher (2 M) or lower (50 mM) than those present physiologically (150 mM). The phase transition at such solvent conditions underscores the complexity of the dynamic processes that underlie condensate formation and disassembly and the factors that influence them, such as fluctuations in the condensate milieu and favored ionic or hydrophobic interactions in non-physiological environments.

## After induction of phase separation by M1-linked ubiquitin chains, NEMO assemblies persist upon polyubiquitin hydrolysis

The foregoing experiments established that M1-linked tetra-ubiquitin is required to induce phase separation of NEMO. To probe whether M1-linked ubiquitin chains are also required for the maintenance of NEMO assemblies, we induced phase transition of GFP-tagged NEMO with M1-linked tetra-ubiquitin and then added recombinant OTULIN to hydrolyze M1-linked ubiquitin chains. The samples were analyzed by SDS–PAGE and Coomassie blue staining to monitor the hydrolysis of M1-linked tetra-ubiquitin chains and by fluorescent microscopy to visualize the presence of NEMO assemblies. First, we induced phase separation of NEMO by TEV-mediated cleavage of MBP-NEMO-GFP in the presence of M1-linked tetra-ubiquitin for 1 h (Fig 4F, lane 7). After adding OTULIN to the NEMO assemblies and incubating for an additional hour, M1-linked

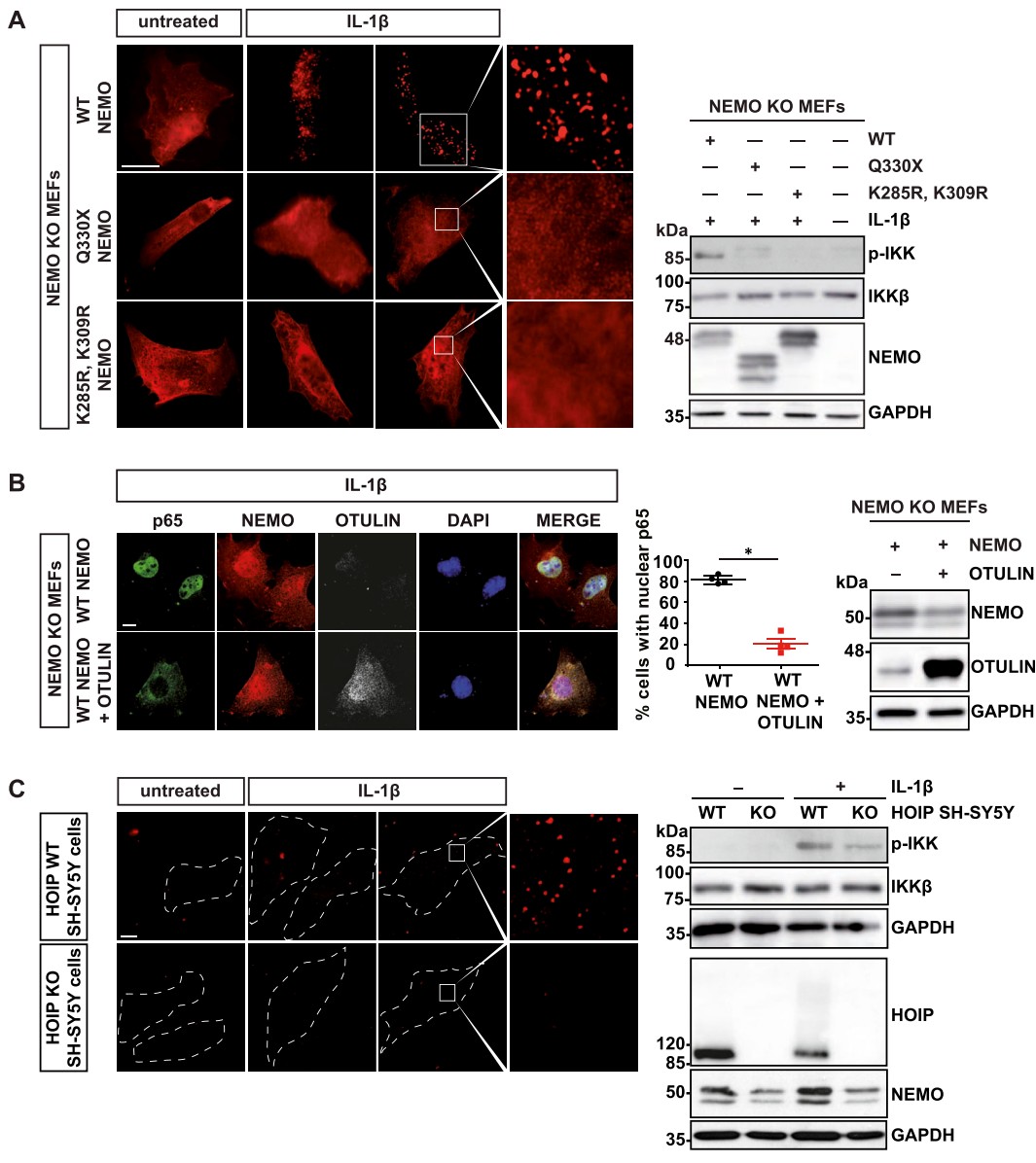

**Figure 2. Linear ubiquitination promotes the formation of NEMO assemblies and IKK complex activation upon IL-1β receptor activation.**
**(A)** In contrast to WT NEMO, Q330X and K285/309R NEMO do not form foci upon IL-1β treatment. NEMO KO MEFs were reconstituted with HA-tagged WT NEMO, Q330X NEMO or K285/309R NEMO. 24 h later, the cells were stimulated with murine IL-1β (20 ng/ml) for 15 min or left untreated and then analyzed after saponin extraction by immunocytochemistry using an antibody against NEMO (scale bar, 10 μm). Right Panel: IKK complex activation was analyzed by immunoblotting using an antibody against phospho-IKKα/β. Antibodies against IKKβ and NEMO were used to control expression levels. GAPDH was immunoblotted as input control. **(B)** IL-1β–induced nuclear translocation of p65 is impaired by OTULIN. The cells were treated as described in (A) and analyzed by immunocytochemistry using antibodies against NEMO, OTULIN, and p65 (scale bar, 10 μm). Right panel: Data are displayed as mean ± SD and were analyzed by Mann–Whitney test, n = 4, *P ≤ 0.05. Expression levels of HA-tagged WT NEMO and OTULIN were analyzed by immunoblotting using antibodies against NEMO and OTULIN. GAPDH was immunoblotted as input control. **(C)** IL-1β–induced formation of NEMO assemblies and IKK complex activation are compromised in HOIP-deficient cells. Control SH-SY5Y cells and HOIP KO SH-SY5Y cells were stimulated with human IL-1β (20 ng/ml) for 10 min and then analyzed by immunocytochemistry after saponin extraction using an antibody against NEMO (scale bar, 10 μm). Right Panel: IKK complex activation was analyzed by immunoblotting using an antibody against phospho-IKKα/β. Antibodies against IKKβ, HOIP, and NEMO were used to control expression. GAPDH was immunoblotted as input control.
Source data are available for this figure.

tetra-ubiquitin chains were hydrolyzed, reflected by the disappearance of the band corresponding to tetra-ubiquitin and the appearance of a new band corresponding to mono-ubiquitin on the Coomassie-stained gel. Yet, the microscopic analysis revealed that NEMO assemblies were still present under these conditions (Fig 4F,

lane 5). These results suggested that heterotypic interactions of NEMO with M1-linked tetra-ubiquitin are essential for condensate formation; however, the persistence of NEMO assemblies might be attributed to homotypic NEMO interactions, at least in vitro and during the time scale of this experiment.

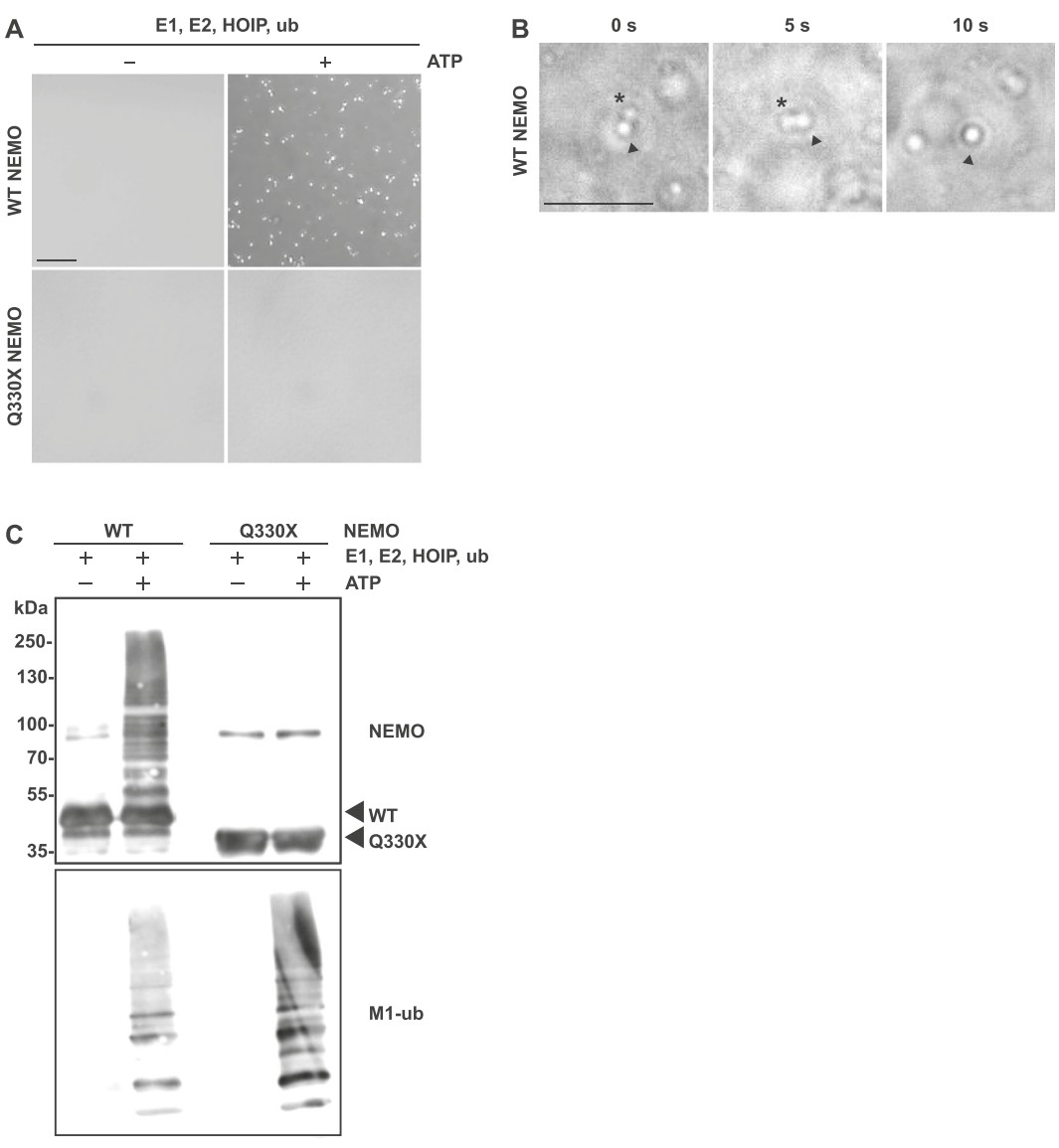

**Figure 3. WT NEMO undergoes phase separation in an M1-ubiquitin–dependent manner.**
**(A)** In contrast to Q330X NEMO, WT NEMO undergoes LLPS upon linear ubiquitination by HOIP. The samples from the in vitro ubiquitination assay described in (C) were analyzed by bright-field microscopy. Representative images are shown (scale bar, 20 $\mu$m). **(B)** Liquid droplets formed by ubiquitinated NEMO are dynamic. WT NEMO ubiquitinated by HOIP as described in (C) were analyzed by bright-field microscopy to follow up fusion events. The images were taken every 5 s over a period of 10 s (scale bar, 20 $\mu$m). **(C)** WT but not Q330X NEMO is modified by M1-linked ubiquitin in vitro. Untagged WT NEMO and Q330X NEMO were subjected to an in vitro linear (M1-linked) ubiquitination assay by incubating the recombinant proteins (5 $\mu$M) with 50 $\mu$M mono-ubiquitin (ub), 1.5 $\mu$M ubiquitin-activating enzyme UBe1, 4 $\mu$M E2 ubiquitin-conjugating enzyme UbcH5c, 4 $\mu$M C-terminal HOIP comprising the RBR and LDD domains, ATP, and MgCl$_2$ in 50 mM HEPES buffer (pH 7.4) containing 0.5 mM TCEP (tris(2-carboxyethyl)phosphine) for 2 h at RT. As a control, ATP was omitted (–ATP). The samples were analyzed by immunoblotting using antibodies against NEMO and M1-ubiquitin.
Source data are available for this figure.

## Binding of Q330X NEMO to M1-linked tetra-ubiquitin is impaired

The Q330X NEMO mutant did not form assemblies in the presence of M1-linked polyubiquitin. Therefore, we compared the binding affinities of WT NEMO and Q330X NEMO with M1-linked tetra-ubiquitin by intrinsic fluorescence spectroscopy. Whereas WT NEMO bound to 4×M1-ub with high affinity ($K_b$ = [5.0 ± 2.5]•10$^6$ M$^{-1}$), consistent with previous reports (Komander et al, 2009; Lo et al, 2009; Rahighi et al, 2009; Yoshikawa et al, 2009), the binding constant between Q330X NEMO and 4×M1-ub was reduced by a factor of 25 ($K_b$ = [0.20 ± 0.04] •10$^6$ M$^{-1}$) (Fig 5A). It is interesting to note that in our binding analysis, a stoichiometry of 4:2 was observed for WT NEMO, indicating that on average one NEMO tetramer can bind two 4×M1-ub molecules. This is in accordance with the 2:1 binding observed for the NEMO UBAN domain in complex with tetra-ubiquitin in solution (Vincendeau et al, 2016; Jussupow et al, 2020). Conversely, for Q330X NEMO, a 1:1

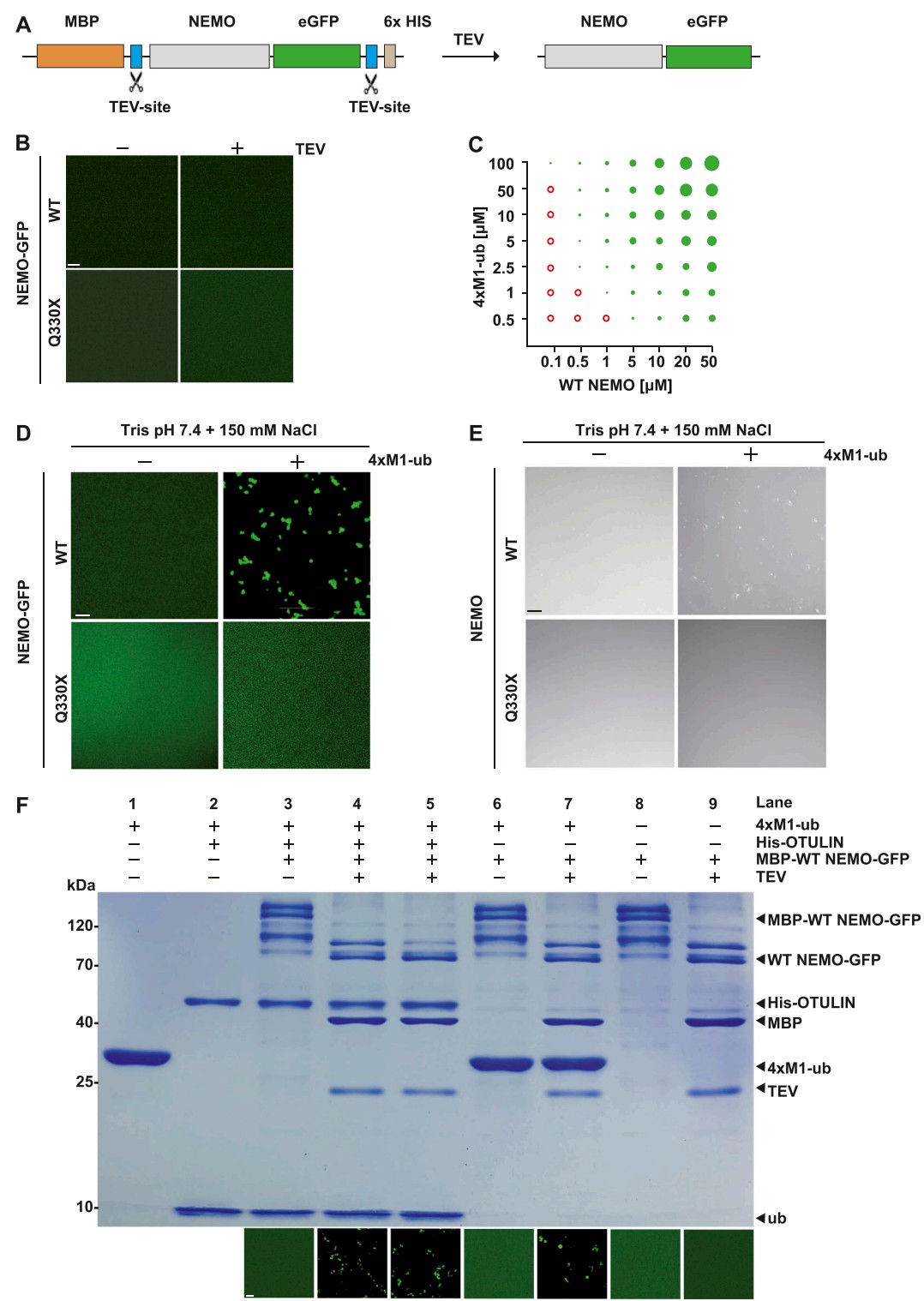

**Figure 4. M1-linked tetra-ubiquitin induces phase transition of NEMO in vitro.**
**(A)** Domain structure of recombinant WT NEMO-GFP or Q330X NEMO-GFP. MBP (maltose-binding protein), TEV (Tobacco Etch Virus protease). **(B)** WT NEMO-GFP and Q330X NEMO-GFP do not phase-separate. Fusion proteins were incubated in the presence of TEV protease (1 h at RT) to cleave off the N-terminal MBP and the C-terminal His₆ tag and then WT NEMO-GFP and Q330X NEMO-GFP were analyzed by fluorescent microscopy using a laser scanning microscope. **(C)** Concentration-dependent phase separation of WT NEMO and M1-linked tetra-ubiquitin (4×M1-ub). WT NEMO-GFP was incubated in presence of recombinant M1-linked tetra-ubiquitin (4×M1-ub) at the concentrations indicated and analyzed by fluorescence microscopy. Red empty circles: no phase separation; green solid circles: phase separation; sizes of the circles illustrate the sizes of liquid droplets. **(D, E)** WT NEMO but not Q330X NEMO undergoes LLPS in the presence of M1-linked tetra-ubiquitin. GFP-tagged (D, E) or untagged (E) WT NEMO and Q330X NEMO (5 µM in 10 mM Tris, pH 7.4, and 150 mM sodium chloride [NaCl]) were incubated in the presence or absence of 10 µM 4×M1-ub and analyzed by

stoichiometry was determined, revealing that one monomer of mutant NEMO binds to only one 4×M1-ub molecule. Notably, Q330X NEMO contains the major M1-ubiquitination acceptor sites (K285 and K309) and an intact M1-ubiquitin–binding UBAN domain (residues 296–327) (Rahighi et al, 2022), but is neither modified by HOIP nor binds to M1-linked polyubiquitin. To find possible reasons for the differences between WT NEMO and Q330X NEMO, we used bioinformatic tools and biophysical approaches to characterize and compare the secondary and ternary structures of WT and mutant NEMO. First, we performed circular dichroism (CD) spectroscopy and small-angle X-ray scattering (SAXS) with recombinant untagged proteins to explore structural differences between WT NEMO and Q330X NEMO. The CD spectra showed that WT NEMO and Q330X NEMO share a similar, predominantly α-helical secondary structure with two well-defined minima at around 223 and 208 nm and a positive band below 200 nm (Fig 5B), which is consistent with previous CD spectra of WT NEMO (Catici et al, 2015). Likewise, SAXS revealed similar X-ray scattering patterns from WT NEMO and Q330X NEMO, reflected by the calculated radius of gyration, $R_g$, values. The $R_g$ of Q330X NEMO of 3.03 ± 0.2 nm seems to be slightly larger than that of WT NEMO ($R_g$ = 2.64 ± 0.2 nm) (Fig 5C) but is still similar except for a small difference in the slope. Thus, CD and SAXS did not provide evidence for marked structural differences between WT NEMO and Q330X NEMO. Of note, high resolution structures of NEMO are only available for NEMO subdomains, with the largest fragment comprising the CC2, UBAN, and LZ domain (https://www.ebi.ac.uk/pdbe/pdbe-kb/proteins/Q9Y6K9/structures/).

### N- and C-terminal domains contribute to M1-ubiquitin–induced phase separation of NEMO

Next, we performed a bioinformatic analysis using predictive algorithms. IUPred, Metapredict, Simple Modular Architecture Research Tool, and ODiNPred revealed intrinsically disordered regions (IDRs) in the N- and C-terminal domains of NEMO as well as three low-complexity domains (LCDs) (residues 251–281, 313–340, and 364–394). Classification of Intrinsically Disordered Ensemble Regions (CIDER) identified NEMO as a weak polyampholyte, based on the fraction of charged residues and net charge per residue, which promotes compaction (Fig 6A). CIDER also categorized NEMO into the family of Janus sequences that show either collapsed or expanded sequences and context-dependent behavior, which is mainly influenced by factors like salt concentration, ligand binding, and interactions. The C-terminal IDR/LCD domains and the ZF domain that are missing in the Q330X NEMO mutant presumably facilitate ubiquitin binding and thus phase separation of NEMO. To address this possibility experimentally, we cloned and expressed NEMO-GFP mutants comprising either the CoZi domain (249–344 aa) or the CoZi domain combined with the C-terminal domain (CoZi-CTD,

249–419 aa). The CoZi domain encompasses the coiled-coil 2 domain, the UBAN domain, and the leucine zipper and has been shown to be necessary for binding to M1-linked and K63-linked ubiquitin chains (Ea et al, 2006; Wu et al, 2006; Rahighi et al, 2009). Interestingly, neither CoZi NEMO nor CoZi-CTD NEMO underwent phase separation when incubated with recombinant M1-linked tetra-ubiquitin, similarly to Q330X NEMO (Fig 6B). Thus, binding to M1-linked ubiquitin is necessary but not sufficient to induce LLPS of NEMO. Both N- and C-terminal domains of NEMO including the dimerization domain, IDRs, and LCDs obviously contribute to homotypic and heterotypic interactions required for inducing its phase separation.

## Discussion

Cellular signaling pathways need to be tightly regulated in a spatiotemporal manner. The formation of biomolecular condensates by phase separation allows for the fast and reversible assembly of signalosomes, which can promote signal amplification by several features: They compartmentalize signaling components, provide a specialized microenvironment, and enhance catalytic reactions by bringing substrates, enzymes, and their regulatory factors in close proximity (Li et al, 2012; Wippich et al, 2013; Hnisz et al, 2015; Su et al, 2016; Du & Chen, 2018; Case et al, 2019). The NF-κB pathway is a well-established paradigm for the effective regulation of signaling by enzymatic posttranslational modifications, such as phosphorylation and ubiquitination (Karin & Lin, 2002; Hayden & Ghosh, 2012; Kanarek & Ben-Neriah, 2012; Napetschnig & Wu, 2013). Here we show that NEMO undergoes phase separation in vitro by binding to linear ubiquitin chains and forms assemblies upon IL-1β receptor activation in a HOIP-dependent manner.

Previous data and our bioinformatic analysis predicted that NEMO has a significant portion of structural disorder (Catici et al, 2015). This substantial degree of native disorder allows NEMO to occupy a wide equilibrium of different conformational states that are separated by relatively low energy barriers and thus have the potential to interconvert rapidly (Catici et al, 2015). We reasoned that these structural features in combination with its potential to oligomerize suggest a propensity of NEMO to undergo phase separation. However, NEMO did not phase separate on its own in a wide range of buffers with different pH conditions. Because binding to M1-linked ubiquitin chains promotes canonical NF-κB activation in response to various stimuli, we tested whether linear ubiquitin chains can induce phase separation of NEMO. In vitro, both M1-linked ubiquitin chains assembled by HOIP and preformed recombinant M1-linked tetra-ubiquitin–induced phase separation of NEMO. Importantly, mono-ubiquitin did not induce phase separation of NEMO, indicating the importance of heterotypic multivalent interactions. The crucial role of linear ubiquitin chains in

fluorescence microscopy (D) or bright-field microscopy (E). Shown are representative images; scale bar, 10 μm. **(F)** Preformed NEMO assemblies persist upon OTULIN-mediated hydrolysis of M1-linked tetra-ubiquitin. MBP-WT NEMO-GFP was incubated with the components indicated (lanes 3–9). In the reaction corresponding to lane 4, recombinant OTULIN was added together with TEV protease, whereas in the reaction corresponding to lane 5 OTULIN was added 1 h after TEV protease. Lane 1: M1-linked tetra-ubiquitin only. Lane 2: M1-linked tetra-ubiquitin incubated with OTULIN for 1 h. The samples were analyzed by SDS–PAGE and Coomassie blue staining (upper panel) and laser scanning microscopy (lower panel).
Source data are available for this figure.

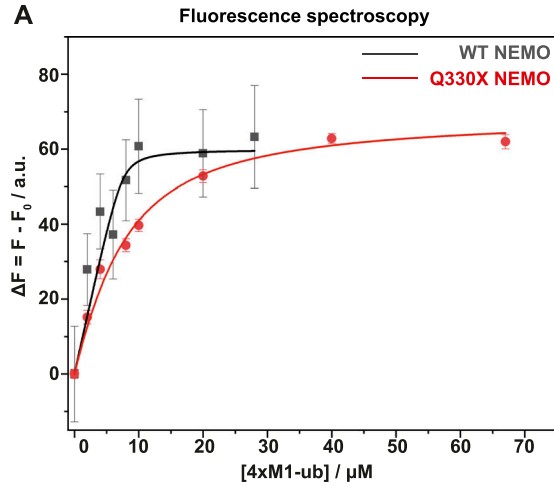

**A** Fluorescence spectroscopy

| proteins | $K_b$ / M$^{-1}$ | n[a] |
|---|---|---|
| WT NEMO / 4xM1-ub | $(5.0\pm2.5)\times10^6$ | 0.5 |
| Q330X NEMO / 4xM1-ub | $(0.20\pm0.04)\times10^6$ | 1 |

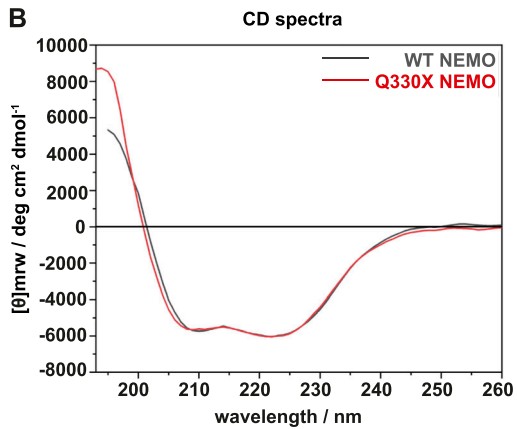

**B** CD spectra

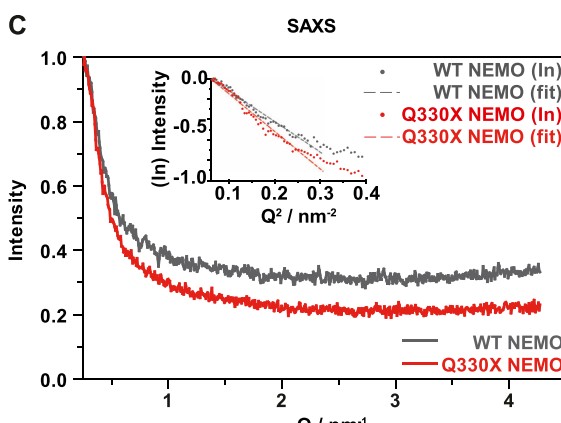

**C** SAXS

| protein | $R_g$ / nm |
|---|---|
| WT NEMO | $(2.64 \pm 0.2)$ |
| Q330X NEMO | $(3.03 \pm 0.2)$ |

**Figure 5. Binding of Q330X NEMO to M1-linked ubiquitin is impaired.**
**(A)** Q330X NEMO shows a reduced binding affinity for 4×M1-ub. Binding isotherms obtained by steady-state fluorescence spectroscopy for the complex formation between untagged WT NEMO (black squares) or Q330X NEMO (red circles) and recombinant 4×M1-ub at 25°C in 10 mM Tris–HCl buffer, pH 7.4. The solid lines represent the best fit of experimental data according to an equivalent and independent binding site model. [a]n is the stoichiometry defined as mol of linear tetra-ubiquitin per mol of WT NEMO or Q330X NEMO. **(B)** WT NEMO and Q330X NEMO show similar, predominantly alpha-helical secondary structures. Circular dichroism spectra were recorded in the Far-UV region (below 260 nm) for WT NEMO and Q330X NEMO at 1 mg/ml concentration. **(C)** WT NEMO and Q330X NEMO show similar conformations in solution reflected by comparable radius of gyration values obtained by small-angle X-ray scattering. The small-angle X-ray scattering measurements of WT NEMO and mutant Q330X NEMO were made at a concentration of 5 mg/ml in 10 mM Tris, pH 7.4. The data were obtained by the program "PRIMUS" and are shown as a function of Guinier linear plot for intensity versus Q.

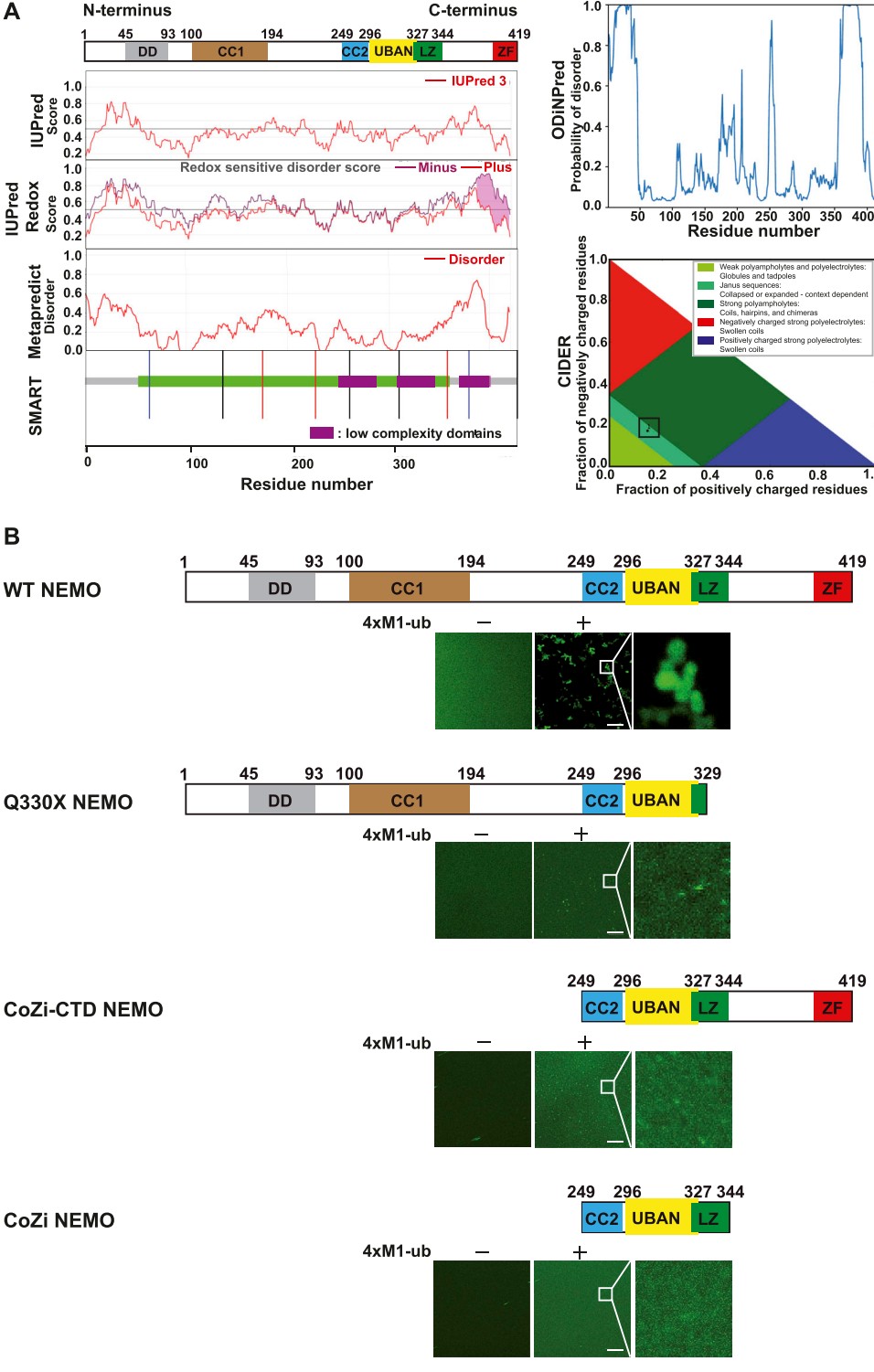

**Figure 6. N- and C-terminal domains contribute to M1-ubiquitin–induced phase separation of NEMO.**
**(A)** Bioinformatic analyses predicts IDRs in the N- and C-terminus of WT NEMO along with spanning low complexity domains. Schematic representation of the human NEMO domain structure is shown on top. DD, dimerization domain; CC, coiled coil; UBAN, ubiquitin binding in ABIN and NEMO; LZ, leucine zipper; ZF, zinc finger. The following bioinformatic tools were used: IUPred, prediction of intrinsically unstructured proteins; SMART, simple modular architecture research tool; ODiNPred, prediction of order and disorder by evaluation of NMR data. Classification of NEMO based on the fraction of charged residues by CIDER: classification of intrinsically disordered ensemble regions. **(B)** Both N- and C-terminal domains are essential for phase separation of NEMO. Fusion proteins (5 μM) composed of MBP and WT NEMO-GFP, Q330X NEMO-GFP or CoZi-CTD NEMO-GFP (aa 249–419), or CoZi NEMO-GFP (aa 249–344) were incubated in presence of TEV protease to cleave off the MBP tag for 1 h at RT in buffer containing 10 mM Tris pH 7.4 ± 10 μM 4×M1-ub and then analyzed by fluorescent microscopy using a laser scanning microscope (scale bar, 10 μm).

inducing NEMO phase separation was further documented by analyzing the pathogenic mutant Q330X NEMO, which is defective in NF-κB activation and linked to the NEMO loss-of-function pathology *Incontinentia pigmenti*. We found that this mutant is characterized by the lack of both covalent ubiquitination by HOIP

and binding to linear ubiquitin chains, explaining why it does not undergo phase separation. Upon IL-1β receptor activation, neither Q330X NEMO nor K285/309R NEMO, another mutant defective in linear ubiquitination, formed assemblies or promoted IKK activation. Moreover, endogenous NEMO was impaired in IL-1β–induced

foci formation and IKK activation in the absence of HOIP, indicating a HOIP-dependent coupling of these events. Increased expression of OTULIN interfered with IL-1β–induced NF-κB activation in cells by depleting M1-linked ubiquitin chains. However, when recombinant OTULIN was added to NEMO assemblies already formed in the presence of M1-linked tetra-ubiquitin in vitro, the NEMO condensates persisted despite OTULIN-mediated hydrolysis of M1-linked tetra-ubiquitin. These observations suggest that M1-linked ubiquitin chains are required for the induction of condensate formation; however, the persistence of NEMO assemblies could be linked to changes in material properties over time. These observations point towards a complex regulation of NEMO condensate dynamics in cells, which requires further investigation.

During the preparation of this manuscript, a similar finding was published by Chen and coworkers (Du et al, 2022). In their study, the authors comprehensively tested the effects of polyubiquitin binding on NEMO phase separation. Du et al showed that binding of NEMO to K63- and M1-linked ubiquitin induces phase separation by multivalent interactions and that IKKα and IKKβ are activated within NEMO condensates. K63-linked polyubiquitin binds to the UBAN domain, albeit with a lower affinity compared with M1-linked polyubiquitin (Lo et al, 2009; Rahighi et al, 2009; Kensche et al, 2012), and to the C-terminal ZF (Laplantine et al, 2009). Consistently, NEMO mutants lacking the UBAN domain or the C-terminal ZF were defective in undergoing phase separation (Du et al, 2022). Interestingly, deletion of the C-terminal ZF prevented M1-linked tetra-ubiquitin–induced phase separation of NEMO, although binding of M1-linked ubiquitin to NEMO obviously does not require the ZF (Laplantine et al, 2009). Thus, the ZF seems to affect NEMO phase separation not directly by binding to M1-linked ubiquitin chains but rather by modulating heterotypic or homotypic interactions. Notably, our mutational analysis indicated that binding to M1-linked ubiquitin is required but not sufficient to induce phase separation of NEMO. A C-terminal NEMO fragment comprising the UBAN and ZF domains did not phase separate in the presence of M1-linked tetra-ubiquitin, suggesting that the N-terminal dimerization domain (by favoring homotypic interactions) and possibly disordered N-terminal regions (by influencing binding affinities or solvent interactions) contribute to phase separation of NEMO.

K63- and M1-ubiquitination–induced transient formation of NEMO punctate structures upon IL-1β stimulation of cells has also been observed by Tarantino et al (2014). By using super-resolution microscopy, Scholefield et al reported that in non-stimulated cells, NEMO forms a supramolecular lattice-like network by binding to K63- and M1-linked polyubiquitin and IKKs (Scholefield et al, 2016). They concluded that these pre-existing lattices are primed to rapidly respond to activating stimuli, such as IL-1β, and facilitate the formation of compact condensates for proximity-enhanced activation. Our data support such a model by providing evidence for a critical role of linear ubiquitin chains in triggering the transition into compact, phase-separated NEMO assemblies. Because the abundance of M1-linked ubiquitin chains under basal conditions is extremely low, a stimulus-induced increase of M1-ubiquitin chain formation is perfectly suited to induce this transition. We recently discovered that M1- and K63-linked ubiquitin chains are formed at the mitochondrial outer membrane in response to TNF and IL-1β stimulation, resulting in IKK complex activation (Wu et al, 2022). Thus, mitochondria may provide a membrane scaffold for

NEMO condensate formation that allows efficient signal amplification based on the large surface of mitochondria. In support of this concept, binding of NEMO to linear polyubiquitin induces and/or selects for a conformational state that promotes membrane association and enhances the affinity for IKKβ (Catici et al, 2015).

Finally, our study provides further evidence that ubiquitin is a potent regulator of phase separation (Dao & Castaneda, 2020; Lei et al, 2020; Sun et al, 2020; Baker & Bernardini, 2021; Luo et al, 2021). This regulatory activity spans a wide spectrum of cellular functions, including p62-mediated autophagy (Sun et al, 2018; Zaffagnini et al, 2018; Turco et al, 2019), UBQLN2- and RAD23B-mediated shuttling of proteins to the proteasome (Dao et al, 2018, 2019), or G3BP1-mediated disassembly of stress granules (Gwon et al, 2021), and adds another layer to the complex but intriguing facets of ubiquitin biology.

# Materials and Methods

### Plasmids

Plasmid maintenance and amplification were carried out using *E. coli* TOP10 (ThermoFisher Scientific) and baculovirus. NEMO constructs were generated by standard PCR cloning techniques and are based on the coding region of human NEMO (GenBank accession number AF091453.1). Codon-optimized NEMO 5′-ATGAAT CGCCATTTGTGGAAATCTCAGCTGTGCGAAATGGTGCAACCGTCTGGTGGAC CAGCGGCGGATCAGGATGTGTTGGGTGAAGAATCGCCATTAGGCAAGCCTG CGATGCTGCATCTGCCGAGCGAACAGGGTGCTCCGGAGACTCTGCAACGCTG CTTAGAGGAAAATCAAGAACTGCGTGATGCGATTCGTCAAAGCAACCAGATT CTGCGCGAACGCTGTGAAGAACTTCTGCACTTTCAGGCATCGCAACGCGAAG AAAAGGAATTTCTGATGTGCAAATTCCAGGAAGCACGTAAACTGGTTGAACG CCTGGGCTTAGAAAAGTTGGACCTGAAACGCCAGAAGGAACAGGCGTTACG TGAAGTCGAACACCTTAAACGCTGTCAGCAGCAAATGGCCGAGGATAAAGCT TCCGTGAAAGCCCAGGTTACGAGCTTACTGGGTGAACTGCAGGAATCCCAGT CACGCCTTGAGGCCGCCACAAAAGAGTGTCAGGCCTTGGAAGGGCGTGCAC GTGCAGCGAGTGAGCAAGCCCGTCAGCTGGAAAGCGAACGTGAAGCCCTGC AACAACAGCATTCGGTCCAAGTGGATCAGCTGCGTATGCAGGGGCAGAGCGT AGAGGCTGCTTTGCGGATGGAACGTCAGGCGGCGTCCGAAGAGAAACGCAA ACTGGCTCAGCTGCAAGTGGCGTATCATCAGCTGTTCCAGGAGTACGACAAC CACATCAAATCAAGCGTTGTAGGCAGTGAGCGCAAACGGGGCATGCAACTGG AGGACCTCAAACAGCAACTCCAACAAGCGGAAGAAGCCCTGGTAGCGAAACA AGAAGTGATTGACAAACTCAAAGAAGAAGCCGAGCAGCATAAGATCGTGATG GAAACCGTTCCGGTCTTAAAAGCCCAGGCAGATATCTACAAAGCTGACTTTCA GGCGGAACGTCAAGCACGCGAAAAGTTAGCGGAAAAGAAAGAGCTGCTGCA AGAGCAGCTGGAGCAGCTGCAGCGCGAGTATTCCAAACTGAAAGCCAGTTGT CAGGAAAGCGCGCGCATTGAGGATATGCGCAAACGCCATGTGGAAGTTTCTC AAGCGCCCCTTCCGCCAGCACCGGCATACCTTTCTAGTCCGCTCGCATTGCCG TCGCAGCGTCGGTCACCTCCGGAAGAGCCTCCGGACTTCTGCTGTCCCAAAT GCCAGTATCAGGCTCCAGATATGGATACCCTCCAGATCCACGTCATGGAATG CATTGAATAA-3′ was ordered from Eurofins (https://www.eurofins. com/). To generate the pMAL-MBP-TEV-NEMO-eGFP-TEV-His$_6$ plasmid, the PrP-coding region was exchanged from the pMAL-MBP-TEV-PrP-eGFP-TEV-His$_6$ plasmid, provided by Janine Kamps (Kamps et al, 2021). WT NEMO: aa 1–419, mutant Q330X NEMO: aa 1–329 (aa 330 changed to stop codon). MBP-WT NEMO-GFP was generated using the following primers: WT NEMO-fwd: 5′-ATAGTCGACATGAATCGCCATTTGTGGAA-3′ and WT NEMO-rev: 5′-

ATAGGATCCTTCAATGCATTCCATGAC-3′. MBP-Q330X NEMO-GFP was generated using the following primers: Q330X NEMO-fwd: 5′-ATAGTCGACATGAATCGCCATTTGTGGAA-3′ and Q330X NEMO-rev: 5′-ATAGGATCCTTGCAGCAGCTCTTTCTTTT-3′. MBP-CoZi-CTD NEMO-GFP was generated using the following primers: CoZi-CTD NEMO-fwd: 5′-ATAGTCGACGTTGTAGGCAGTGAGCGCA-3′ and CoZi-CTD NEMO-rev: 5′-ATAGGATCCTTCAATGCATTCCATGACGTGGA-3′. MBP-CoZi NEMO-GFP was generated using the following primers: CoZi NEMO-fwd: 5′-ATAGTCGACGTTGTAGGCAGTGAGCGCAA-3′ and CoZi NEMO-rev: 5′-ATAGGATCCTTTCAGTTTGGAATACTCGC-3′. The amplified fragments were digested with SalI and BamHI and cloned into the pMAL-MBP-TEV-PrP-eGFP-TEV-His$_6$ plasmid.

For expression in insect cells, human WT NEMO protein (Uniprot ID Q9Y6K9) or Q330X NEMO was sub-cloned into pOPIN-His-MBP multi-host expression vectors (https://www.ncbi.nlm.nih.gov/pmc/articles/PMC1874605/) (Berrow et al, 2007). The MBP fusion could be removed by cleavage with 3C protease. The sequence was amplified via PCR and recombination sites were added through the primer. MBP-WT NEMO was generated using the following primers: WT NEMO-fwd: 5′-GAAGTTCTGTTTCAGGGTCCCAATCGCCATTTGTGGAAA TCTCAGCT-3′ and WT NEMO-rev: 5′-TAAACTGGTCTAGAAAGCTTATT-CAATGCATTCCATGACGTGGATCTGG-3′. MBP-Q330X NEMO was generated using the following primers: Q330X NEMO-fwd: 5′-GAAG TTCTGTTTCAGGGTCCCAATCGCCATTTGTGGAAATCTCAGCT-3′ and Q330X NEMO-rev: 5′-TAAACTGGTCTAGAAAGCTTACAGCAGCTCTTTCTTTTCCGC TAACT-3′. The pOPIN-His-MBP backbone was linearized by digestion with restriction enzymes KpnI and HindIII and purified by agarose gel electrophoresis. The PCR product and vector were assembled via SLIC by incubation first with T4 Polymerase in the absence of dNTPs for 10 min at 22°C and second with recombinase A (New England Biolabs) in the presence of dCTP for 30 min at 37°C, followed by immediate transformation into E. coli OmniMAX competent cells. Positive clones were screened by colony PCR, followed by Sanger sequencing of the whole ORF.

The recombinant 4×M1-ub chains were constructed synthetically with four ubiquitin moieties whose DNA sequence was codon optimized. This synthetic gene was then cloned into pGEX-2rbs vector provided by Andrea Mussacchio. pET28b His-mouseUBE1 was obtained from Addgene (#32534). pET47b-HOIP-RBR-LDD (C-terminal HOIP) and pGEX-UbcH5c were provided by Katrin Rittinger (Stieglitz et al, 2012), and pcDNA 3.1 (−) WT OTULIN was provided by Daniel Krappmann (Stangl et al, 2019).

HA-tagged WT NEMO and K285/309R NEMO were sub-cloned using the following primers: HA-NEMO-fwd: 5′-ATATGGATCCAA-TAGGCACCTCTGGAAGAGC-3′, HA-NEMO-rev: 5′-ATAT GCGGCCGCCTA CTCAATGCACTCCATGACATG-3′ from FLAG-tagged WT NEMO and FLAG-tagged K285/309R NEMO respectively (Muller-Rischart et al, 2013). HA-tagged Q330X NEMO was generated using the following primers: HA-Q330X NEMO-fwd: 5′-ATATGGATCCAATAGGCACCTCTGGA AGAGC-3′, HA-Q330X NEMO-rev: 5′-ATATGCGGCCGCTCACAGGAGCT CCTTCTTCTCGG-3′. The amplified fragment was digested with BamHI and NotI and cloned into pcDNA3.1-N-HA.

## Cell culture

MEFs derived from NEMO KO mice (Schmidt-Supprian et al, 2000) and HEK 293T cells (CRL-1573; American Type Culture Collection) were cultured in DMEM supplemented with 10% (vol/vol) FBS, 100 IU/ml penicillin, and 100 µg/ml streptomycin sulfate. SH-SY5Y cells (DSMZ no. ACC 209) were cultured in DMEM F-12 (DMEM/F12) supplemented with 15% (vol/vol) FBS, 100 IU/ml penicillin, 100 µg/ml streptomycin sulfate, and 1 × MEM non-essential amino acids solution (Gibco).

## Generation of HOIP CRISPR/Cas9 KO SH-SY5Y cells

sgRNAs (RNF31-24147982 AGGGUGUUGAGGUAGUUUCG; RNF31-24147993 GAGCCGUGGACAGGGUGUUG) were designed using the Synthego website (http://design.synthego.com). 1.5 nmol sgRNAs were rehydrated in 50 µl nuclease-free 1 × TE buffer (10 mM Tris–HCl, 1 mM EDTA, pH 8.0) to a final concentration of 30 µM (30 pmol/µl). sgRNA and recombinant CAS9 were delivered as RNP complexes using a 4D-Nucleofector X-Unit (Lonza). Briefly, for the assembly of the RNP complexes, Cas9 2NLS and sgRNAs were combined in Nucleofector solution at a molar ratio of 9:1 sgRNA to Cas9 and incubated for 10 min at room temperature. The cells were resus-pended at a concentration of 150,000 cells/5 µl. 5 µl of the cell suspension was added to the 25 µl of pre-complexed RNPs for a total transfection volume of 30 µl per reaction and transferred to Nucleofector cartridges. Nucleofection was performed according to the predefined protocol (CA-137 for SH-SY5Y cells), and cells were carefully resuspended in each well of the Nucleocuvette with 70 µl of pre-warmed growth medium and transferred to the pre-warmed six-well and incubated in a humidified 37°C/5% CO$_2$ incubator. After 24 h the medium was replaced.

For clone screening, the cells were split into two six-well cell culture plates, and pools were analyzed by PCR and subsequent DNA sequencing. For this, primer pairs (HOIP_fwd: AGTCC-CACCCTCTCTCCTAG HOIP_rev: TGTGACTGTAGCAACCTGGT) were ordered extending ~200–250 bp 3′ and 5′ of the sgRNA binding region. To perform cell pool or single clone sequencing analysis, genomic DNA was isolated using a genomic DNA extraction kit (Monarch Genomic DNA Purification Kit; New England Biolabs), and the PCR was optimized to yield a single amplicon. After PCR product pu-rification (NucleoSpin Gel and PCR Clean-up; Macherey-Nagel GmbH), the DNA was sent for Sanger sequence analysis (Micro-synth Seqlab GmbH). The KO efficiency of the cell pools and single colony clones was determined using the Synthego ICE analysis website (https://ice.synthego.com). To isolate single KO clones, the KO cell pools were diluted to 1 cell/100 µl and 5 cells/100 µl and the dilutions were distributed over several 96-well plates. 15–25 clones were grown from single cells and reanalyzed using the above-mentioned process. Finally, clones with a high KO score were amplified, and KO efficiency was confirmed by immunoblotting.

## Recombinant protein expression and purification from *E. coli*

pMAL-MBP-WT_NEMO-eGFP constructs were transformed into *E. coli* Rossetta (pLysS) host strain. For protein expression, 4 liter bacterial culture of lysogeny broth medium was inoculated and grown to an OD (600 nm) of 0.6–1.0. Expression was induced with 300 µM IPTG, and culture was incubated overnight at 25°C at 120 rpm. For MBP-Q330X_NEMO-eGFP, MBP-CoZi-CTD_NEMO-eGFP, and MBP-CoZi_NEMO-eGFP, expression was induced with 300 µM IPTG and incubated over night at 16°C, 120 rpm. Bacteria were harvested

by centrifugation (5,000*g*, 4°C, 20 min), and the pellets were washed with 20 ml PBS and centrifuged again (4,000*g*, 4°C, 20 min). Pellets were stored at −80°C until further use. For purification, the bacterial pellet was resuspended in lysis buffer (50 mM Tris pH 8.0, 250 mM NaCl, 10% glycerol, 2 mM β-mercaptoethanol, DNase, protease inhibitor). Protein lysis was performed via SLM AMINCO French Press (Thermo Fisher Scientific), and the protein solution was centrifuged (40,000*g*, 45 min, 4°C). The supernatant was diluted in a 1:1 ratio and loaded on a His-Trap FF column (GE Healthcare) equilibrated with a lysis buffer and washed with three column volume lysis buffer containing 20 mM imidazole. Proteins were eluted with lysis buffer containing 500 mM imidazole and dialyzed over night at 4°C in dialysis buffer (50 mM Tris pH 8.0, 25 mM NaCl, 2 mM β-mercaptoethanol). The protein was then loaded on an ion exchange column and purified with a gradient of salt concentration from 25 mM to 2 M NaCl. The final fraction corresponding to the correct molecular weight of MBP-WT_NEMO-eGFP (121,188 D) and MBP-Q330X_NEMO-eGFP (111,010 kD) was aliquoted and flash frozen until further use.

The purification of mouse UBE1 was performed as described previously (Carvalho et al, 2012). The purification of HOIP-RBR-LDD (C-terminal HOIP) and UbcH5c was carried out as described previously (Stieglitz et al, 2012).

For the purification of recombinant linear tetra-ubiquitin chain, pGEX-4×M1-ub was freshly transformed into BL21 (DE3) pLysS cells. With help of pre-culture grown at 25°C overnight, the cells were grown in TB medium at 37°C till the OD reaches 0.8–1. 0. After cooling down to 20°C, induction was initiated with 0.5 mM IPTG and then grown overnight. Harvested cells were resuspended in a lysis buffer (20 mM Tris pH 8.0, 250 mM NaCl, 5% glycerol, 4 mM BME) supplemented with a protease inhibitor cocktail. The resuspended cells were lysed using a French press and then centrifuged (40,000 rcf, 45 min, 4°C) to pellet cell debris. The supernatant was then loaded onto the GSTrap FF column (Cytiva). After a thorough wash, the GST-4×M1-ub was eluted with 20 mM glutathione in lysis buffer. The elute was treated with PreScission protease for the cleavage of the GST tag and dialyzed in a buffer containing low salt (20 mM Tris pH 8.0, 25 mM NaCl, 1 mM DTT). This sample was loaded onto CaptoQ ImpRes (Cytica). This step trapped GST and other impurities, while 4×M1-ub did not bind. The flow-through corresponding to the correct molecular weight of 4×M1-ub (34,205 D) was exchanged for a storage buffer (20 mM HEPES pH 8.0, 150 mM NaCl, 1 mM TCEP) and flash frozen until further use.

### Recombinant protein expression and purification from Sf9 and High Five insect cells

Expression tests in insect cells were performed to identify the optimal conditions. His-MBP-WT NEMO WT was expressed in Sf9 cells, while His-MBP-Q330X NEMO was expressed in High Five cells. Both proteins were expressed on a 2-liter scale. A culture was started in three sterile 2-liter flasks, each with $3.0 \times 10^6$ (Sf9) and $1.5 \times 10^6$ (High Five) cells/ml from mid-log phase suspension stock cultures. The cells were grown at 27°C and 120 rpm and infected with 21 ml (Sf9) or 11 ml (High Five) P3 baculovirus (amplified from test expression culture) for each flask (63 or 33 ml total). Cells were incubated for 48 h (Sf9; WT NEMO) or 72 h (High Five; Q330X NEMO). After taking aliquots of all cultures for analysis, the culture was

harvested in 1-liter centrifuge beakers (3,000 rcf, 15 min, 4°C). The supernatant was removed, and the pellet was washed twice with cold PBS by centrifugation (3,900 rcf for 15 min at 4°C).

The cell pellets from the 2-liter expression cultures were resuspended in 5 ml lysis buffer per gram pellet (50 mM HEPES, 300 mM NaCl, 10 mM imidazole, 1 mM TCEP, pH 8.0) with 5 U/ml benzonase, AEBSF, and 0.2% Triton X-100. The resuspended cells were lysed using a Cell Disrupter TS 0.75 (Constant Systems) at 1 kilobar. Lysates were centrifuged at 75,000 rcf at 10°C, and the supernatant was filtered with 3 μm filters. The lysate was purified using an Äkta Xpress System (Cytiva, former GE Healthcare) with a 5 ml Ni-NTA affinity column, followed directly by a S75 26/60 size exclusion column. The protein was digested on column with 3 ml HRV-3C protease (made in house, 4 mg/ml; 200 U/mg) for 5 h at 4°C to remove the purification tag. The on-column cleavage was successful. The elution fractions, which appeared to contain the most protein, were pooled, centrifuged, and loaded onto the gel filtration (GF) column. The GF column was eluted with 1.2 column volumes of GF buffer (25 mM HEPES, 40 mM, NaCl, 1 mM TCEP, pH 8.0). The eluted peaks were collected in the fraction collector plate and analyzed by capillary electrophoresis.

For WT NEMO, the SEC peak corresponding with the size of the cleaved protein (48,221 D) was collected and the fractions pooled. Likewise, the SEC peak of Q330X NEMO (37,915 D) was pooled. The final fraction was aliquoted and flash frozen until further use. The yield for WT NEMO was 15.2 ml with 1.04 mg/ml (21.47 μM) for a total of 15.81 mg from 2-liter expression culture. The yield for NEMO Q330X was 15.3 ml with 1.67 mg/ml (44.13 μM) for a total of 25.55 mg from 2-liter expression culture.

### Immunocytochemistry

MEFs or SH-SY5Y cells were cultivated on glass coverslips (Laboratory Glassware Marienfeld). 24 h after transfection, cells were treated with 20 ng/ml murine or human IL-1β (Thermo Fisher Scientific). To analyze puncta formation, saponin extraction was performed before fixation. For this purpose, cells were first washed twice with ice-cold PBS and then treated twice with ice-cold saponin extraction buffer (80 mM Pipes, pH 6.8, 1 mM MgCl₂, 1 mM EGTA, and 0.1% saponin) for 2 and 4 min on ice for MEFs and 2 min on ice for SH-SY5Y cells. Then the cells were washed twice with PBS before fixation. All steps were performed on ice with ice-cold buffers and then fixed for 15 min with 4% paraformaldehyde in PBS, pH 7.4, and permeabilized with 0.2% (vol/vol) Triton X-100 in PBS for 10 min plus 5% goat/donkey serum in PBS for 45 min at room temperature. After blocking with 5% goat/donkey serum, cells were stained with primary antibodies at a dilution of 1:250 in blocking solution at 4°C overnight, washed with PBS, and incubated with fluorescent dye-conjugated secondary antibodies, Alexa Fluor 488 (Thermo Fisher Scientific), at a dilution of 1:1,000 for 1 h at room temperature. Cells were mounted in Fluoroshield and imaged by confocal super-resolution microscopy.

### Nuclear translocation of p65

NEMO KO MEFs were cultivated on glass coverslips (Laboratory Glassware Marienfeld). After 24 h, cells were transiently transfected

with 1 µg of HA-tagged WT NEMO or Q330X NEMO. In case of co-transfection, 0.5 µg of each HA-tagged WT NEMO and OTULIN were used. 24 h after transfection, cells were treated for 15 min with 20 ng/ml murine IL-1 (Thermo Fisher Scientific). All steps were performed on ice, with ice-cold buffers, and then fixed for 15 min with 4% paraformaldehyde in PBS, pH 7.4, permeabilized with 0.2% (vol/vol) Triton X-100 in PBS for 10 min, and blocked with 5% goat/donkey serum and 1% BSA in 0.2% (vol/vol) Triton X-100 in PBS for 45 min at room temperature. After blocking, cells were stained with primary antibodies against HA tag or NEMO, OTULIN, and p65 at a dilution of 1:100–1:500 in blocking solution at 4°C overnight, washed with PBS, and incubated with fluorescent dye-conjugated secondary antibodies Alexa Fluor 488, 555, and 647 (Thermo Fisher Scientific), at a dilution of 1:500–1:1,000 for 1 h at room temperature. Cells were mounted in Fluoroshield with DAPI (Sigma-Aldrich) and analyzed by SR-SIM.

## Quantification and statistical analysis

After preparation of the samples on coverslips, they were analyzed using the green, red, and DAPI channel of a fluorescence microscope. Cells containing nuclear green (anti-p65) staining were counted positive for nuclear p65 translocation, and only cells positive for the transfected construct were considered. For quantification, three to five independent experiments were performed, and at least 200 cells were analyzed per condition collectively. Statistical analysis was carried out using the Kruskal-Wallis test followed by Dunn's Multiple Comparison Test or nonparametric $t$ test and the Mann–Whitney test to determine significant differences between samples (significance levels: $*P \leq 0.05$; $**P \leq 0.01$; $***P \leq 0.001$).

## Immunoprecipitation

Cells were lysed in 50 mM HEPES, pH 7.4, buffer containing 150 mM NaCl, 1 mM EDTA, 1 mM EGTA, 1% Triton, 10% glycerol, 25 mM NaF, 10 µM ZnCl2 in PBS supplemented with 30 mM NEM (Sigma-Aldrich), protease inhibitor (complete; Roche), and phosphatase inhibitor (PhosStop; Roche). The lysates were cleared by centrifugation (20,000$g$, 4°C, 20 min). Samples were incubated for 4 h with anti-HA agarose beads (Thermo Fisher Scientific) at 4°C under rotation. Beads were washed five times with the same buffer. Immuno-purified proteins were eluted by adding 4× Laemmli sample buffer and boiling for 10 min at 95°C. Samples were analyzed by immunoblotting using the indicated antibodies.

## Immunoblotting

Proteins were size-fractionated by SDS–PAGE (10% polyacrylamide) and transferred to nitrocellulose or PVDF at 0.45 µm by electro-blotting. The membranes were blocked with 5% non-fat dry milk or 5% BSA in TBST (TBS containing 0.1% Tween 20) for 30 min at room temperature and subsequently incubated with the primary antibody at a dilution of 1:1,000 in the same blocking solution for overnight at 4°C or for 1 h at 37°C. After extensive washing with TBST, the membranes were incubated with horseradish peroxidase-conjugated secondary antibody at a dilution of 1:5,000 in TBST

for 1 h at room temperature. After washing with TBST, the antigen was detected with the enhanced chemiluminescence detection system (Amersham Biosciences) as specified by the manufacturer and imaged using X-ray films or an Azure Sapphire Biomolecular Imager (Azure Biosystems).

## Confocal laser scanning microscopy

Fluorescent imaging laser scanning microscopy was performed using an LSM880 (Carl Zeiss) with a 63× oil immersion objective. For Z-stack scanning a 63× NA 1.4 oil immersion objective was used to record a stack of 67.5 × 67.5 × 10 and 0.330 µm for each optical section. The argon laser power was set to 0.006% at 488 nm with pixel dwell time of 5.71 µs. During all measurements, laser power, gain, and field of view were kept constant. The Z-stacks were then processed to obtain maximum intensity projections. Data were imported into Imaris 9.3.1 for three-dimensional analysis of the Z-stack images, and the surface module was used for reconstruction of the surfaces.

## SR-SIM

An ELYRA PS.1 microscope (Carl Zeiss) with a 63× oil immersion objective was used for structured-illumination microscopy (SIM) with five phases and three to five rotations of the SIM grid and acquisition times of 100–150 ms. All channels were acquired independently and subsequently. Raw confocal SIM images were processed and generated using the ZEN Black 2.1 software (Carl Zeiss).

## Sample preparation

Protein aliquots were thawed on ice and centrifuged at 20,000$g$ for 10 min at 4°C to remove aggregates. Using Vivaspin 500 columns with 30 or 10 kD molecular weight cut off (Sartorius Stedim Biotech), the buffer was exchanged to 10 mM Tris, pH 7.4, unless mentioned otherwise by centrifuging five to eight times for 9 min at 12,000$g$ and 4°C. After buffer exchange, the final protein concentration was determined by NanoDrop 2000. To study phase transition, TEV protease was added to the sample and incubated 1 h for complete cleavage before microscopy in case of GFP-tagged NEMO. 5 µM of NEMO was used for all the in vitro phase separation experiments (unless specifically mentioned otherwise). After the reaction, 10 µl of reaction mix was spotted on ibidi coverslip bottom dishes.

## Linear ubiquitination assay

50 µM mono-ubiquitin (ub), 1.5 µM ubiquitin-activating enzyme UBe1, 4 µM E2 ubiquitin-conjugating enzyme UbcH5c, and 4 µM C-terminal HOIP comprising the RBR and LDD domains were incubated with WT NEMO or Q330X NEMO plus/minus ATP/MgCl$_2$ in 50 mM HEPES buffer, pH 7.4, containing 0.5 mM TCEP (tris(2-carboxyethyl) phosphine) for 2 h at room (25°C) or body (37°C) temperature. A few µl of the sample were directly viewed using bright-field microscopy, and the reaction was stopped thereafter by adding Laemmli sample buffer and boiling for 10 min and then analyzed by immunoblotting using M1-ubiquitin-specific

antibodies ($1 \times 10^3$; Millipore) and NEMO-specific antibodies from Sigma-Aldrich.

## Deubiquitination assay

10 $\mu$M M1-linked tetra-ubiquitin (4×M1-ub) and 2 $\mu$M OTULIN were incubated either with 5 $\mu$M MBP-WT NEMO-GFP and TEV in 10 mM Tris buffer, pH 7.4, containing 1 mM DTT (dithiothreitol) for 1 h at room (25°C) temperature OR for 1 h post-TEV cleavage of MBP-WT NEMO-GFP. A few $\mu$l of the sample were directly viewed using confocal laser scanning microscopy, and the reaction was stopped thereafter by adding Laemmli sample buffer and boiling for 10 min and then analyzed by SDS–PAGE and Coomassie blue staining.

## Fluorescence spectroscopy

The interaction between WT NEMO and mutant Q330X NEMO with linear tetra-ubiquitin was monitored by steady-state fluorescence spectroscopy by using a K2 spectrofluorometer by ISS (Champaign). The cuvette path-length was 0.3 cm. The temperature was set at 25°C and it was controlled by a circulating water bath directly connected to the sample holder. Briefly, a series of solution at fixed concentrations of NEMO (15 $\mu$M) and Q330X NEMO (5 $\mu$M) were prepared. Instead, the concentration of linear tetra-ubiquitin was varied between 0 and 30 $\mu$M (for titration with NEMO) and 70 $\mu$M (for titration with Q330X NEMO). The extent of binding was monitored by exciting the samples at 295 nm and collecting the fluorescence emission at 333 nm, the maximum of emission of NEMO and Q330X NEMO. In order to obtain the value of the binding constant, $K_b$, which describes the association process between the interacting partners, and the stoichiometry, $n$, defined as the moles of 4×M1-ub bound per mole of protein, plots of $\Delta F = F - F_0$ (where $F$ and $F_0$ are the fluorescence intensities in the presence and absence of the binding partner, respectively) versus total tetra-ubiquitin concentration were produced. The experimental data were fitted with an equivalent and independent binding site model, as described previously in detail (Oliva et al, 2020).

## CD spectroscopy

CD spectra were recorded in the Far-UV region (in the range 195–260 nm) by means of a Jasco J-715 spectropolarimeter (from Jasco Corporation) and using a quartz cuvette with a path-length of 0.1 cm. The temperature was set at 25°C by means of a circulating water bath directly connected to the sample holder. The concentrations of NEMO and Q330X NEMO were 1.9 and 2.4 $\mu$M, respectively. The following instrument parameters were used: scan rate of 50 nm min$^{-1}$, band width of 5 nm, and response time of 2 s. For each sample, a background (10 mM Tris–HCl buffer, pH 7.4) was recorded and subtracted from the sample. The final spectra are the results of three accumulations. The spectra were normalized per mole of residues and are, thus, reported as mean molar ellipticity.

## SAXS

The SAXS measurements were performed by a SAXSess mc$^2$ instrument from Anton-Paar. The temperature was kept constant at 20°C with the help of a TCS Control Unit (Anton-Paar). The X-ray radiation from an X-ray tube made of copper was directed onto a quartz capillary ($\mu$-cell; Anton-Paar) filled with 10 $\mu$l of a 5 wt% protein solution. The scattering for each sample was detected via imaging plates for 30 min in an evacuated chamber to reduce air scattering. Analysis of the intensity ($I$) versus momentum transfer ($Q = 4\pi\sin\theta/\lambda$; $\lambda$ = wavelength of radiation, $2\theta$ = scattering angle) data and calculation of $R_g$ values were carried out using the PRIMUS software (Konarev et al, 2003).

## Bioinformatic analysis

IUPred2 (https://iupred2a.elte.hu/), Metapredict (https://metapredict.net/), and ODiNPred (https://st-protein.chem.au.dk/odinpred) were used to predict IDRs. A Simple Modular Architecture Research Tool (http://smart.embl-heidelberg.de/) was used to define and locate LCDs. CIDER (http://pappulab.wustl.edu/CIDER/) was used to categorise based on charge parameters.

### Antibodies

| Antibody | Source | Identifier |
| --- | --- | --- |
| Rabbit monoclonal anti-HA | Cell Signaling Technology | Cat# 3724 |
| Rabbit polyclonal anti-IKBKG/NEMO | Sigma-Aldrich | Cat# HPA000426, RRID: AB_1851572 |
| Rabbit monoclonal anti-linear ubiquitin | Millipore | Cat# MABS199, RRID: AB_2576212 |
| Mouse monoclonal anti $\beta$-actin | Sigma-Aldrich | Cat# A5316, RRID: AB_476743 |
| Rabbit monoclonal anti-NF-$\kappa$B p65 | Cell Signaling Technology | Cat# 8242, RRID: AB_10859369 |
| Rabbit polyclonal anti-HOIP | Bethyl Laboratories | Cat# A303-560A, RRID: AB_10949139 |
| Rabbit polyclonal anti-I$\kappa$B$\alpha$ | Cell Signaling Technology | Cat# 9242 |
| Rabbit monoclonal anti-IKK$\beta$ | Cell Signaling Technology | Cat# 8943 |
| Rabbit polyclonal anti-phospho-IKK $\alpha/\beta$ (Ser176, Ser180) | Thermo Fisher Scientific | Cat# 710676, RRID: AB_2532752 |
| Mouse monoclonal anti-GAPDH | Calbiochem | Cat# CB1001 |
| Rabbit polyclonal anti-OTULIN | Cell Signaling Technology | Cat# 14127 |

# Supplementary Information

# Acknowledgements

We thank Karin Rittinger and Ben Stieglitz for the HOIP-RBR-LDD and UbcH5c plasmids; Daniel Krappmann for the OTULIN plasmid; Janine Kamps for the pMAL-MBP-TEV-PrP-eGFP-TEV-His6 plasmid; Andrea Musacchio for PGEX-2rbs plasmid; Lena Angersbach, Nikolas Furthmann, and Sandy Kokoschka for HA-tagged WT NEMO, Q330X NEMO, and K285/309R NEMO plasmids; Tina Gazdag and Stefanie Dorok for cloning and expression of untagged NEMO proteins in insect cells; Sanjib Mukherjee for help with fluorescence spectroscopy; and Marc Schmidt-Supprian for NEMO-KO MEFs. KF Winklhofer is supported by the German Research Foundation (WI/2111-4, WI/2111-6, WI/2111/8, FOR 2848, and Germany's Excellence Strategy – EXC 2033 – 390677874 – RESOLV) and the Michael J Fox Foundation for Parkinson's Research (Grant ID 16293). SR-SIM microscopy was funded by the German Research Foundation and the State Government of North Rhine-Westphalia (INST 213/840-1 FUGG).

## Author Contributions

S Goel: conceptualization, data curation, formal analysis, validation, investigation, visualization, methodology, and writing—original draft.

R Oliva: resources, formal analysis, validation, investigation, methodology, and writing—review and editing.

S Jeganathan: resources and methodology.

V Bader, LJ Krause, and ID Stender: resources, investigation, and methodology.

S Kriegler: investigation and methodology.

CW Christine and K Nakamura: resources and writing—review and editing.

J-E Hoffmann: resources, supervision, validation, methodology, and writing—review and editing.

R Winter: conceptualization, supervision, validation, project administration, and writing—review and editing.

J Tatzelt: conceptualization, supervision, validation, visualization, writing—original draft, and project administration.

KF Winklhofer: conceptualization, supervision, funding acquisition, validation, visualization, writing—original draft, and project administration.

## Conflict of Interest Statement

The authors declare that they have no conflict of interest.

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
