## [Reviewer comments · Life Science Alliance]

Life Science Alliance

Linear ubiquitination induces NEMO phase separation to activate NF- κ B signaling

Simran Goel, Rosario Oliva, Sadasivam Jeganathan, Verian Bader, Laura Krause, Simon Kriegler, Isabelle Stender, Chadwick Christine, Ken Nakamura, Jan-Erik Hoffmann, Roland Winter, Jörg Tatzelt, and Konstanze Winklhofer

DOI: <https://doi.org/10.26508/lsa.202201607>

Corresponding author(s): Konstanze Winklhofer, Ruhr University Bochum

Review Timeline:	Submission Date:	2022-07-15
	Editorial Decision:	2022-07-19
	Revision Received:	2022-11-01
	Editorial Decision:	2022-11-21
	Revision Received:	2022-12-22
	Editorial Decision:	2023-01-03
	Revision Received:	2023-01-09
	Accepted:	2023-01-09

Scientific Editor: Novella Guidi

Transaction Report:

Please note that the manuscript was previously reviewed at another journal and the reports were taken into account in the decision-making process at Life Science Alliance.

Referee #1 Review

Report for Author:

In the present study, Winklhofer and colleagues demonstrate that purified NEMO is soluble but forms large cluster, which can be visualized by bright-field and fluorescence microscopy, upon incubation with the E3 ligase HOIP that conjugates Met1-linked ubiquitin chains. These assemblies are indicative of phase separation and are not observed with a patient-derived NEMO truncation mutant. Biophysical analyses suggest that intrinsically disordered regions in the NEMO C-terminus are required for NEMO clustering and IKK/NF- κ B activation. Immune fluorescence analyses show that ubiquitin-triggered NEMO assemblies can be detected upon overexpression in MEFs in response to IL-1 β stimulation and that clustering correlates with NF- κ B activation.

Even though the manuscript provides some interesting starting points, it lacks an in depth analyses on the phenomenon of phase separation by NEMO/ubiquitin assemblies and its role for activation of the NEMO/IKK complex in cells. Most data seem somewhat premature and no statistical analyses are included. Obviously, the manuscript was wrapped up very rapidly, because a comprehensive analyses on this topic has recently been published by the Chen laboratory in Molecular Cell (Du et al, preprint 26th of April 2022). The authors mention the paper in the discussion and it would certainly be an option to publish a second parallel and independent study that comes to similar conclusions. However, the major problem is that the present study cannot stand on its own without the thorough analyses performed by the Chen lab. They have provided a very rigorous analysis in 7 main figures and 16 supplementary figures that involved extensive quantified imaging, time-resolved fluorescence and FRAP analyses using two stimulations, alternative cellular purifications approaches, comprehensive mutagenesis of NEMO including a number of patient-derived mutations, and clear correlations to cellular IKK activation. There is not a single aspect of the present manuscript that goes beyond the initial paper and overall it reports only on initial observations. I think the manuscript would need to add a new aspect to justify considering it for publication, e.g. the composition of the cellular clusters, the potential cooperativity of different E3 ligases and NEMO ubiquitin binding and conjugation, or the functions of DUBs in counteracting the assemblies post-stimulation.

Some specific comments:

Statistical analyses are missing throughout.

Is the C-terminal fusion of EGFP to NEMO impacting on NEMO function and potentially phase separation? An IP-derived NEMO stop codon mutation leading to a C-terminal extension impacts on NEMO function (PMID: 10839543). Thus, the effect of C-terminal NEMO fusions must be accurately controlled.

Many panels contain only negative results that could be moved to a supplement (e.g. Figure 1C, 1D, 3). Especially the inclusion of data from Figure 3 is not clear. Binding affinities of Ub chains to NEMO have been extensively studied (also +/- ZnF), SAXS results for NEMO have been published (e.g. PMID: 33055165), and the comparison of NEMO WT and Q330X does not provide relevant insights for the regulation of phase separation. It is known that NEMO Q330X cannot rescue NF- κ B signaling and no additional evidence is presented that disordered regions in the C-terminus are controlling phase separation. Thus, the figure is not well connected to the story and could be removed or moved to the supplement.

Referee #2 Review

Report for Author:

Goel et al. aims to examine how linear polyubiquitin chains, either through direct modification of NEMO or noncovalent binding to NEMO, promotes phase separation of NEMO. Furthermore, the authors postulate that the interactions of these ubiquitin chains with NEMO is linked to NEMO function in activating NF- κ B. As currently written, this study largely mirrors the findings of very recent observations published in *Molecular Cell* on 4/26 by a different group (Du et al. 2022, DOI: 10.1016/j.molcel.2022.03.037). The current manuscript also appears incomplete, as many questions are left unanswered or results are left inconclusive in the text, particularly with the mutant studies in Figure 3 and the cell-based work in Figure 4. For these reasons, I do not currently support publication of this work in this journal. The authors need to identify other aspects of their work that complement or distinguish their work from Du et al. 2022, but that is not readily apparent in the current version.

In the Du et al. 2022 work, the authors examined effects of different polyubiquitin chains (all linkages) and showed that either K63 or M1-linked chains interacted with NEMO to drive NEMO condensation by heterotypic interactions. In addition, that study also examined contributions of various Ub-binding domains (ZF) to the ability of NEMO to undergo heterotypic phase separation with polyubiquitin chains. In these two observations, the Du et al. work is more comprehensive than the current work, which used only M1-linked polyUb chains and presented suggestive but inconclusive data regarding the NEMO Q330X mutant. At this time, the new contribution by the current work is that linear ubiquitination of NEMO appears to also promote NEMO phase separation, however this result is also impacted by the presence of free linear polyubiquitin chains in Figure 1. I list major concerns below to improve the manuscript, potentially in a submission to a different journal.

Major Comments

1) Figure 1, as presented, is written in such a manner as if the authors wanted to find a condition under which NEMO phase separated. The authors use a variety of conditions to show that the protein does not phase separate on its own, but rather requires polyubiquitin chains (only shown under one condition in Figure 1F). A stronger start would have emphasized that NEMO phase separates with polyubiquitin chains and then dissected these contributions.

2) Figure 2: These data are minimal, and potentially best incorporated with Figure 1 or partially as a supplementary figure. The authors could have examined NEMO phase separation under different conditions (temperatures, NEMO concentrations, and M1-Ub4 to NEMO ratios) to better quantify phase-separating conditions. It is becoming more routine to at least construct a phase diagram as a function of a variable (such as protein concentration or ligand ratio). Further discussion on the material properties of these NEMO/polyUb clusters is also lacking, but important given their low sphericity and amorphous characteristics. For these reasons, these studies are considered underdeveloped.

3) The data regarding the Q330X NEMO mutant are perplexing and inconclusive:

- Presumably, the C-terminal zinc finger domain is important for Ub binding, thus deletion of this domain in the Q330X mutant would significantly alter ability of this mutant to interact with polyUb. Indeed, there are other publications that point to the Ub-binding properties of ZF (as the authors point out, such as Laplantine et al. 2009), although these studies appear to suggest that the ZF domain is K63-specific. Given that there are multiple domains within NEMO that interact with Ub, it is important to dissect how avidity contributes to polyubiquitin/NEMO interactions, and how polyubiquitin linkage contributes to this. Unfortunately, none of this is discussed in the current manuscript.

- On a related manner, the SAXS data for wild-type and the Q330X mutant are confusing. In the text, the authors suggest that the Rg increases quite a bit for the Q330X mutant (by 4 angstroms compared to WT), although this doesn't appear to be the case in the table in Figure 3C). The Guinier plot for the Q330X is also not linear. How do the SAXS data contribute to an understanding of the effects of Q330X on the structure of NEMO?

- Finally, a discussion on the oligomerization state of NEMO is necessary given the postulated importance of oligomerization to NEMO's ability to phase separate. It is briefly mentioned in the text that NEMO tetramerizes. Is oligomerization affected in the Q330X mutant?

4) From what I can tell, no data are shown to illustrate that the NEMO assemblies in cells are signaling-competent; therefore, the

title "IL1 β induces the formation of signaling-competent NEMO assemblies in an M1-ubiquitin-dependent manner" on pg. 7 is very preliminary. Furthermore, all work in Figure 4 used NEMO knockout MEF cells. Given that these are transient transfection experiments to introduce WT NEMO or mutant NEMO, it is important to control for NEMO overexpression and to compare NEMO expression levels with endogenous NEMO expression levels. Additionally, the images in Figure 4 are representative; no statistics are provided for the number of cells analyzed, puncta count, etc. The OTULIN DUB overexpression work is correlative, but other controls need to be performed (with other DUBs, e.g. K63-selective DUBs) to strengthen the connections among NEMO condensation, polyubiquitination, and NF-KB activation.

5) The following sentence sends an incorrect message regarding the role of intrinsically-disordered regions (IDR) in driving phase separation: "Our bioinformatic analysis predicted that NEMO consists of 22% intrinsically disordered regions. Almost 50% of the protein are likely to be disordered, and 41% of the secondary structure are random coil. Altogether, these structural features suggest a propensity of NEMO to undergo phase separation." There have been many recent reviews and research papers that are de-emphasizing the connection that an IDR drives phase separation. There are many other factors that are important for this consideration, including oligomerization, heterotypic interactions with other proteins to promote phase separation, etc. So, I would not emphasize this IDR connection unless the authors plan to identify the regions within the IDR that are driving phase separation of the NEMO system.

Referee #3 Review

Report for Author:

In this study, Goel et al. report that NEMO forms phase-separated condensates when ubiquitinated by HOIP or when binding to M1-Ub chains in vitro. The authors propose that this phase separation facilitates the activation of NF-kappaB in cells after IL-1 β treatment. To investigate this, the authors compare the properties of WT NEMO and a NEMO truncation mutant (Q330X). The Q330X mutant is defective in M1-Ub chain binding and is not ubiquitinated by HOIP in vitro.

In vitro, phase separation of WT NEMO (but not Q330X NEMO) is claimed based on formation of macromolecular NEMO assemblies facilitated by M1-Ub (conjugated to NEMO or bound by NEMO). In cells, NEMO is shown to form puncta IL-1 β treatment which coincides with nuclear translocation of p65. Puncta formation and p65 translocation does not occur when NEMO KO cells ectopically express NEMO Q330X. The authors conclude that M1-Ub chains facilitate phase separation of NEMO, and this is prerequisite for IKK activation and NF-kB signaling.

Phase separation as a thermodynamic force driving assembly of macromolecular complexes and signal transduction is conceptually very interesting and timely topic to study. However, the current study falls short of demonstrating this, both in the in vitro experiments and in the context cellular signalling (IL-1 β signaling)

It is unclear how the authors distinguish between formation of NEMO: M1-Ub chain complexes that are phase-separated versus non-phase-separated complexes. Detailed measurements of the dynamics of complex assembly should be analysed to determine if indeed phase separation is driving the observed formation of assemblies. Also, the authors do not address that the NEMO puncta in cells actually reflect phase-separation of NEMO, that these puncta represent functional signalling assemblies, or that the phase-separation of prerequisite for NF-kB activation. Hence, it is unclear what is the experimental basis for the author's conclusion of the study that "Linear ubiquitination induces NEMO phase separation to activate NF-kB signaling".

A recent study (Du et al. Mol Cell 2022) reported very similar findings relating to phase separation of NEMO through binding of Ub chains (K63- and M1-Ub). The in vitro experiments of the current study are consistent with this notion but the cellular data is preliminary and should be substantially expanded before any claims regarding NEMO phase separation in cells should be made.

This reviewer is thus left with the notion that the study is quite preliminary and requires substantial development of the cellular assays before any conclusions regarding phase-separation of NEMO by M1-Ub can be made. Conceptually, it will be crucial to demonstrate a) that NEMO puncta in cells represent phase-separated assemblies that are distinct from "normal" macromolecular protein complexes and b) that phase-separation is functionally important for downstream signalling outcomes.

Main comments

The authors provide circumstantial in vitro evidence for phase separation of NEMO in the presence of M1-Ub chains, which is dependent on NEMO's Ub-binding domains. As mentioned in the general comments, the in vitro dynamics of formation of NEMO: M1-Ub chain assemblies needs to be further analysed in detail in order to claim phase-separation. It would also be important to add in additional NEMO signaling complex components - e.g. IKK α /b LUBAC in the experiments to further establish if phase-separation is the thermodynamic force driving formation of the macromolecular assembly of a NEMO signaling complex.

It is well-described that NEMO is recruited to macromolecular receptor signaling complexes (e.g. TNFR1) where it binds K63- and M1-linked polyubiquitin to activate IKK. It is thus not surprising that mutating the Ub-binding domains interferes with the accumulation of NEMO in puncta and IKK/NF-kappaB activation. Indeed, previous studies of NEMO mutations showed that the D311N mutation interfered with M1-ubiquitin binding and NF-kappaB activation (Rahighi et al. 2009). It would be important to address if the detected NEMO puncta indeed represent phase-separated NEMO and, if so, how these puncta are distinct from receptor signalling complexes to which NEMO is recruited? For example, does phase separation occur only when M1-Ub chains reach a certain length. Is phase separation a unique feature of M1-Ub binding or would K63-Ub binding cause induce phase separation. Importantly, it needs to be established that phase separation is required for downstream signaling. Mutating the Ub-binding domain in NEMO will not address this as it also interferes with the recruitment of NEMO to signalling complexes (Tarantino et al. JCB 2014).

Tarantino et al. (JCB 2014) reported that TNF and IL-1b induce rapid re-localisation of NEMO-IKK into supramolecular structures where IKK is activated. They further showed that these structures contain components of the receptor signalling complex, including K63-linked ubiquitin and M1-linked ubiquitin that both contribute to form the supramolecular signaling structure. While the current study refers to these structures as phase separated structures, it seems that the two studies are investigating the same phenomenon. Indeed, Tarantino et al. concluded that the recruitment of NEMO to the puncta was mediated by K63- and M1-linked ubiquitin. The study by Tarantino et al. also provided evidence that the NEMO puncta after TNF are associated the receptor complex whereas the NEMO puncta after IL-1b contain IRAK1 but are not associated with the receptor. Is this also the case in this study? Finally, Tarantino et al. found that NEMO is not required for puncta formation of other components although it is needed for IKK activation. Would this not suggest that phase-separation occurs upstream of NEMO and that the Ub-binding domains of NEMO facilitate the recruitment of NEMO into the structures?

Specific comments

It is stated that modification of NEMO by M1-Ub is well-established. However, several studies have failed to detect M1-linked Ub chains on NEMO following immune receptor stimulation, including after IL-1beta treatment. E.g. see Emmerich et al. PNAS 2013, Fiil et al. Mol Cell 2013. The authors should establish if endogenous/exogenous NEMO indeed is M1-Ub-modified after IL-1beta treatment of their cells.

Related to this, it is stated in the discussion that "Ubiquitination of NEMO by HOIP is required for NF- κ B activation in response to various stimuli." I don't think there is experimental evidence that ubiquitination of endogenous NEMO is functionally important for NF-kappaB activation. If there are studies demonstrating this, the specific studies and stimuli should be indicated.

The use of the Q330X NEMO mutant complicates the interpretation of the data given that the truncation interferes with Ub-binding and NEMO ubiquitination. It would be important to demonstrate that the truncated protein in cells maintains other interaction such interaction with IKKa/IKKb.

In Figure 1 and 2 it is suggested that ubiquitination of NEMO by HOIP in vitro promoted phase separation of NEMO. Since NEMO Q330X is not ubiquitinated by HOIP, the most straight forward interpretation is that HOIP ubiquitinates NEMO on residues within the truncated region and not the ub sites (K285, K309) reported in cells. It would be important to establish that the HOIP-induced NEMO ubiquitination in vitro reflects the ubiquitination of NEMO in cells (with regard to NEMO K residues and Ub chain length). Without this, it seems like a "leap of faith" to claim that in vitro ubiquitination of NEMO and promotion of phase-separation reflects the role of NEMO ubiquitination in response to receptor activation in cells.

July 19, 2022

Re: Life Science Alliance manuscript #LSA-2022-01607-T

Prof. Konstanze F Winklhofer
Ruhr University Bochum
Molecular Cell Biology
Universitaetsstrasse 150
Bochum, North Rhine-Westphalia 44801
Germany

Dear Dr. Winklhofer,

Thank you for submitting your manuscript entitled "Linear ubiquitination induces NEMO phase separation to activate NF- κ B signaling" to Life Science Alliance. The manuscript was assessed by expert reviewers at another journal and then transferred to LSA. We would like to invite further consideration of this manuscript at LSA pending the following revisions:

- Address reviewer 1 specific comments.
- Address Reviewer 2 major comments #1 and #2. Remove inconclusive data on the Q330X NEMO mutant for point #3. Tone down conclusions via discussion for points #4 and #5.
- Address Reviewer 3 specific comment to establish if endogenous/exogenous NEMO is M1-Ub-modified after IL-1beta treatment. Address all the other specific comments via discussion.

Thank you for this interesting contribution to Life Science Alliance. We are looking forward to receiving your revised manuscript.

Sincerely,

B. MANUSCRIPT ORGANIZATION AND FORMATTING:

We would like to thank the referees for their helpful suggestions and constructive criticism. In the following, we addressed their concerns point by point.

The revised version includes substantial new data and information:

1. We quantified the fraction of cells with nuclear 65 translocation upon IL-1 β treatment in NEMO knockout (KO) mouse embryonic fibroblasts (MEFs) reconstituted with either wildtype NEMO or Q330X NEMO (new Fig. 1B) and in NEMO KO MEFs expressing wildtype NEMO with and without OTULIN co-expression (new Fig. 2B).
2. To further strengthen our observation that Q330X NEMO is defective in mediating NF- κ B activation, we analyzed degradation of I κ B α upon IL-1 β receptor activation in cell expressing wildtype NEMO or Q330X NEMO by immunoblotting. This analysis revealed that I κ B α degradation is impaired in cells expressing Q330X NEMO (new Fig. 1C).
3. We included co-immunoprecipitation experiments showing that both wildtype NEMO and Q330X NEMO interact with IKK β , but only wildtype NEMO interacts with HOIP, explaining why Q330X NEMO is not covalently modified by M1-linked ubiquitin chains (new Fig. 1D).
4. We show that the NEMO condensates formed upon HOIP-mediated generation of M1-linked ubiquitin chains are dynamic. Brightfield microscopy revealed that the droplets formed by untagged wildtype NEMO undergo fusion events (new Fig. 3B).
5. A phase diagram illustrates that phase separation of wildtype NEMO induced by M1-linked tetra-ubiquitin is dependent on the concentration of both NEMO and M1-linked ubiquitin (new Fig. 4C). With increasing concentrations of NEMO and M1-linked tetra-ubiquitin larger condensates formed.
6. We also included data revealing that once NEMO phase separation is induced by M1-linked ubiquitin, the NEMO assemblies persist in the presence of OTULIN, which hydrolyzes M1-linked ubiquitin chains (new Fig. 4F). These results suggested that heterotypic interactions of NEMO with M1-linked tetra-ubiquitin are essential for condensate formation, but homotypic NEMO interactions are obviously sufficient to maintain condensates, at least *in vitro* and during the time scale of this experiment.
7. In the revised version, we show that NEMO mutants lacking the N-terminal dimerization domain (DD) and coiled coil 1 (CC1) domain do not phase separate in the presence of M1-linked ubiquitin, although the ubiquitin-binding domains (UBAN, ZF) are present (new Fig.

6B). These data suggest that heterotypic interactions between NEMO and polyubiquitin are not sufficient to induce phase separation of NEMO and that homotypic interactions between NEMO are required in addition.

8. We included size exclusion chromatography profiles of wildtype and Q330X untagged NEMO, showing that their oligomerization profiles are similar (new Supplementary Fig. S1B).

Point-by-Point response

Referee #1

Specific comments

Statistical analyses are missing throughout.

In the revised manuscript, we quantified p65 nuclear translocation and included the statistical analysis (new Figs. 1B and 2B).

Is the C-terminal fusion of EGFP to NEMO impacting on NEMO function and potentially phase separation? An IP-derived NEMO stop codon mutation leading to a C-terminal extension impacts on NEMO function (PMID: 10839543). Thus, the effect of C-terminal NEMO fusions must be accurately controlled.

In our study, we employed both EGFP-tagged NEMO purified from bacteria and untagged NEMO purified from insect cells for a comparative analysis to rule out potential effects of the C-terminal EGFP fusion (Fig. 4D, 4E; Supplementary Fig. S2B-D). Importantly, untagged NEMO was used for the *in vitro* linear ubiquitination assay (Fig. 3A-C) and all the biophysical experiments including fluorescence spectroscopy, circular dichroism and SAXS (Fig. 5A-C).

Many panels contain only negative results that could be moved to a supplement (e.g. Figure 1C, 1D, 3). Especially the inclusion of data from Figure 3 is not clear.

The previous Fig. 1C and 1D have been moved to the Supplement (new Fig. S2A, B). Fig. 3A (new Fig. 5A) demonstrates differences between wildtype NEMO and Q330X NEMO in their binding to M1-linked tetra-ubiquitin regarding their affinity and stoichiometry. The fact that there are no obvious differences in the secondary structure of wildtype NEMO and Q330X NEMO although the binding to M1-linked tetra-ubiquitin is different, is quite interesting and points towards a role of the disordered regions in NEMO. Thus, we would like to keep these panels in the main figures.

The previous Fig. 3D and E (new Fig. 6A) may help to explain why NEMO has the propensity to undergo phase separation, which is informative in light of our new data on NEMO mutants lacking N-terminal domains, which do not phase separate in the presence of M1-linked ubiquitin (new Fig. 6B).

Binding affinities of Ub chains to NEMO have been extensively studied (also +/- ZnF), SAXS results for NEMO have been published (e.g. PMID: 33055165), and the comparison of NEMO WT and Q330X does not provide relevant insights for the regulation of phase separation.

In the publication mentioned by the reviewer, only a fragment of human NEMO (aa 258-350) comprising the UBAN domain was studied, similarly to other studies (Rahighi et al., 2009, Yoshikawa et al., 2009). In our study, we used full-length wildtype NEMO and Q330X NEMO. Notably, ubiquitin binding studies with Q330X NEMO have not been done before. Since phase separation of wildtype NEMO is induced by M1-linked ubiquitin in contrast to Q330X, we feel that our data showing that M1-linked ubiquitin binding to Q330X is impaired are not irrelevant.

It is known that NEMO Q330X cannot rescue NF- κ B signaling and no additional evidence is presented that disordered regions in the C-terminus are controlling phase separation. Thus, the figure is not well connected to the story and could be removed or moved to the supplement.

Q330X NEMO is a novel unpublished mutation we identified and characterized in detail in the accompanying manuscript that has been made available to the reviewers. Thus, it has not been known before that this mutation cannot rescue NF- κ B signaling. In the revised version, we included a functional characterization of this NEMO mutant (new Fig. 1B-D).

We would like to add that in the revised version we included two NEMO mutants comprising the CoZi domain (CC2-UBAN-LZ, aa 249-344) and the CoZi-CTD (CC2-UBAN-LZ-ZF, aa 249-419). Interestingly, neither CoZi nor CoZi-CTD phase separates in the presence of M1-linked ubiquitin, suggesting that ubiquitin binding is required but not sufficient for phase separation of NEMO.

Reviewer #2

Major comments

1) Figure 1, as presented, is written in such a manner as if the authors wanted to find a condition under which NEMO phase separated. Authors rather wanted to test if NEMO could intrinsically undergo phase separation or requires cofactors/ ligands to assist phase transitioning, being a major ubiquitin binding protein. The authors use a variety of conditions to show that the protein does not phase separate on its own, but rather requires polyubiquitin chains (only shown under one condition in Figure 1F). A stronger start would have

emphasized that NEMO phase separates with polyubiquitin chains and then dissected these contributions.

We thank the reviewer for this helpful suggestion. In the revised version, we have rearranged the figures and included more data in order to develop a conclusive storyline. We start with the functional characterization of the pathogenic Q330X NEMO in cells, showing that the Q330X NEMO mutant is defective in IL-1 β signaling and binding to M1-linked polyubiquitin (new Fig. 1). We then introduce the formation of cellular NEMO foci in response to IL-1 β receptor activation (new Fig. 2), before analyzing NEMO condensate formation *in vitro* (Fig. 3, 4, 5 and 6). Thus, the flow of the manuscript is more logical and actually reflects the development of the project.

2) Figure 2: These data are minimal, and potentially best incorporated with Figure 1 or partially as a supplementary figure. The authors could have examined NEMO phase separation under different conditions (temperatures, NEMO concentrations, and M1-Ub4 to NEMO ratios) to better quantify phase-separating conditions. It is becoming more routine to at least construct a phase diagram as a function of a variable (such as protein concentration or ligand ratio). Further discussion on the material properties of these NEMO/polyUb clusters is also lacking, but important given their low sphericity and amorphous characteristics. For these reasons, these studies are considered underdeveloped.

We fully agree with the reviewer. In the revised version, we included a phase diagram for different concentrations of both NEMO and M1-linked tetra-ubiquitin (new Fig. 4C). Regarding material properties, we showed fusion of untagged wildtype NEMO condensates induced by HOIP-mediated linear ubiquitin chain formation by bright-field microscopy (new Fig. 3B).

Previous Fig. 2A and 2B have been moved to the Supplement (new Supplementary Fig. S2C and S2D). Previous Fig. 2C and D have been moved to new Fig. 4D and 4E.

3) The data regarding the Q330X NEMO mutant are perplexing and inconclusive:
- Presumably, the C-terminal zinc finger domain is important for Ub binding, thus deletion of this domain in the Q330X mutant would significantly alter ability of this mutant to interact with polyUb. Indeed, there are other publications that point to the Ub-binding properties of ZF (as the authors point out, such as Laplantine et al. 2009), although these studies appear to suggest that the ZF domain is K63-specific. Given that there are multiple domains within NEMO that interact with Ub, it is important to dissect how avidity contributes to polyubiquitin/NEMO interactions, and how polyubiquitin linkage contributes to this. Unfortunately, none of this is discussed in the current manuscript.

Since the UBAN domain of NEMO domain shows a 100-fold higher affinity for M1-linked ubiquitin compared to K63-linked ubiquitin (Ivins et al., 2009, Komander et al., 2009, Lo et al., 2009, Rahighi et al., 2009), we concentrated in our study on M1-linked ubiquitin. Moreover, binding of M1-linked di-ubiquitin to NEMO is sufficient to activate NF- κ B to full extent in cells, whereas binding of even longer K63-linked ubiquitin chains to NEMO results only in partial activation of NF- κ B (Kensche et al., 2012).

The ZF domain does not contribute to M1-ubiquitin binding (Laplantine et al., 2009), suggesting that the defective phase separation of NEMO- Δ ZF in the presence of M1-linked ubiquitin observed by Du et al. (Du et al., 2022) is based on features other than ZF-mediated ubiquitin binding, which may relate to the presence of low complexity domains in the close vicinity of the ZF. We discussed this issue in our revised version.

- On a related manner, the SAXS data for wild-type and the Q330X mutant are confusing. In the text, the authors suggest that the R_g increases quite a bit for the Q330X mutant (by 4 angstroms compared to WT), although this doesn't appear to be the case in the table in Figure 3C). The Guinier plot for the Q330X is also not linear. How do the SAXS data contribute to an understanding of the effects of Q330X on the structure of NEMO?

Unfortunately, the table in the original Fig. 3C was not updated with data from additional replicates. Those data have now been included in the new Fig. 5C. The data quoted in the results section are correct and correspond to the updated Fig. 5C. SAXS revealed similar X-ray scattering patterns from wildtype NEMO and Q330X NEMO, reflected by the calculated radius of gyration, R_g , values. The R_g of Q330X NEMO of 3.03 ± 0.2 nm seems to be slightly larger than that of WT NEMO ($R_g = 2.64 \pm 0.2$ nm, Fig. 3C), but is still similar except a small difference in the slope. Since this difference is not at very low Q values, it does not affect the Guinier approximation (which contains only some of the first low Q values). In this case, a small difference in the R_g values has no effect on the actual radius in the molecule, as long as the protein structure is spherical.

- Finally, a discussion on the oligomerization state of NEMO is necessary given the postulated importance of oligomerization to NEMO's ability to phase separate. It is briefly mentioned in the text that NEMO tetramerizes. Is oligomerization affected in the Q330X mutant?

In our revised manuscript, we included size exclusion chromatography profiles of recombinant untagged wildtype NEMO and Q330X NEMO purified from insect cells (new Supplementary Fig. S1B), indicating that the oligomerization patterns of wildtype and Q330X NEMO are similar.

4) From what I can tell, no data are shown to illustrate that the NEMO assemblies in cells are signaling-competent; therefore, the title "IL1 β induces the formation of signaling-competent NEMO assemblies in an M1-ubiquitin-dependent manner" on pg. 7 is very preliminary. Furthermore, all work in Figure 4 used NEMO knockout MEF cells. Given that these are transient transfection experiments to introduce WT NEMO or mutant NEMO, it is important to control for NEMO overexpression and to compare NEMO expression levels with endogenous NEMO expression levels. Additionally, the images in Figure 4 are representative; no statistics are provided for the number of cells analyzed, puncta count, etc. The OTULIN DUB overexpression work is correlative, but other controls need to be performed (with other DUBs, e.g. K63-selective DUBs) to strengthen the connections among NEMO condensation, polyubiquitination, and NF-KB activation.

At this stage, we can only correlate NEMO condensate formation with signaling competence in our cellular models. In contrast to wildtype NEMO, the loss-of-function mutant Q330X NEMO does not form condensates *in vitro* and in cells and it does not promote IL-1 β -induced I κ B α degradation or p65 nuclear translocation (new Fig. 1B, 1C, 2A, 3A, 4D, 4E). For the expression of wildtype and mutant NEMO in a NEMO-deficient background (NEMO KO MEFs), we used the minimal amount of plasmids necessary to detect NEMO by immunocytochemistry and immunoblotting. Of note, we always compared the effects of wildtype NEMO and Q330X NEMO and therefore made sure that the expression levels are the same (new Fig. 1C, 1D, 2A).

In the revised version, we quantified the fraction of cells with p65 nuclear translocation (Fig. 1B, 2B). Since we concentrated in our study on HOIP-mediated M1-linked ubiquitin chains, we did not include K63-specific DUBs. This has been done in the study published by Du et al. (Du et al., 2022).

5) The following sentence sends an incorrect message regarding the role of intrinsically-disordered regions (IDR) in driving phase separation: "Our bioinformatic analysis predicted that NEMO consists of 22% intrinsically disordered regions. Almost 50% of the protein are likely to be disordered, and 41% of the secondary structure are random coil. Altogether, these structural features suggest a propensity of NEMO to undergo phase separation." There have been many recent reviews and research papers that are de-emphasizing the connection that an IDR drives phase separation. There are many other factors that are important for this consideration, including oligomerization, heterotypic interactions with other proteins to promote phase separation, etc. So, I would not emphasize this IDR connection unless the authors plan to identify the regions within the IDR that are driving phase separation of the NEMO system.

Yes, we agree with the reviewer. Heterotypic interactions between NEMO and polyubiquitin seem to be a major driving force of NEMO phase separation. However, our new data indicated that polyubiquitin binding is not sufficient to induce phase separation of NEMO, since a NEMO mutant comprising all ubiquitin binding domains (CoZi-CTD, aa 249-419) does not phase separate in the presence of M1-linked tetra-ubiquitin (new Fig. 6B). This suggests that the N-terminal dimerization domain (by favoring homotypic interactions) and possibly disordered N-terminal regions (by influencing binding affinities) contribute to phase separation of NEMO.

Reviewer #3

Main comments

The authors provide circumstantial *in vitro* evidence for phase separation of NEMO in the presence of M1-Ub chains, which is dependent on NEMO's Ub-binding domains. As mentioned in the general comments, the *in vitro* dynamics of formation of NEMO: M1-Ub chain assemblies needs to be further analyzed in detail in order to claim phase-separation. It would also be important to add in additional NEMO signaling complex components - e.g. IKKa/b LUBAC in the experiments to further establish if phase-separation is the thermodynamic force driving formation of the macromolecular assembly of a NEMO signaling complex.

In the *in vitro* experiment shown in Fig. 6A-C, phase separation of NEMO was induced in the presence of HOIP. This mimics a more physiologically relevant situation, since HOIP covalently modifies NEMO with M1-linked ubiquitin chains in addition to generating free M1-linked ubiquitin chains, which bind to the UBAN domain of NEMO.

In the revised manuscript, we analyzed phase separation of NEMO in more detail. We provided evidence for fusion events of condensates formed by untagged full-length NEMO (new Fig. 6B) and included a phase diagram showing the concentration dependency of NEMO phase separation induced by M1-linked tetra-ubiquitin (new Fig. 4C).

It is well-described that NEMO is recruited to macromolecular receptor signaling complexes (e.g. TNFR1) where it binds K63- and M1-linked polyubiquitin to activate IKK. It is thus not surprising that mutating the Ub-binding domains interferes with the accumulation of NEMO in puncta and IKK/NF-kappaB activation. Indeed, previous studies of NEMO mutations showed that the D311N mutation interfered with M1-ubiquitin binding and NF-kappaB activation (Rahighi et al. 2009). It would be important to address if the detected NEMO puncta indeed represent phase-separated NEMO and, if so, how these puncta are distinct from receptor signalling complexes to which NEMO is recruited? For example, does phase separation occur only when M1-Ub chains reach a certain length. Is phase separation a unique feature

of M1-Ub binding or would K63-Ub binding cause induce phase separation. Importantly, it needs to be established that phase separation is required for downstream signaling. Mutating the Ub-binding domain in NEMO will not address this as it also interferes with the recruitment of NEMO to signalling complexes (Tarantino et al. JCB 2014).

We would like to point out that in the Q330X NEMO mutant the UBAN domain (aa 296-327, (Rahighi et al., 2022), required for binding to M1-linked ubiquitin chains, is not affected, whereas the D311N mutation, mentioned by the reviewer, is located within the UBAN domain. The Q330X NEMO mutant lacks the C-terminal ZF domain, which binds K63-linked ubiquitin, but not M1-linked ubiquitin (Laplantine et al., 2009). Thus, the ZF seems to affect NEMO phase separation not directly by binding to M1-linked ubiquitin chains, but rather by modulating heterotypic or homotypic interactions. We focused our study on M1-linked ubiquitin, because binding/conjugation of M1-linked di-ubiquitin to NEMO is sufficient to activate NF- κ B to full extent in cells, whereas binding of even longer K63-linked ubiquitin chains to NEMO results only in partial activation of NF- κ B (Kensche et al., 2012).

A correlative analysis of NEMO puncta formation and signaling competence has already been provided by Du et al. (Du et al., 2022). This study also addressed the role of K63-linked ubiquitin chains and the influence of the length of polyubiquitin chains.

Tarantino et al. (JCB 2014) reported that TNF and IL-1 β induce rapid re-localisation of NEMO-IKK into supramolecular structures where IKK is activated. They further showed that these structures contain components of the receptor signalling complex, including K63-linked ubiquitin and M1-linked ubiquitin that both contribute to form the supramolecular signaling structure. While the current study refers to these structures as phase separated structures, it seems that the two studies are investigating the same phenomenon. Indeed, Tarantino et al. concluded that the recruitment of NEMO to the puncta was mediated by K63- and M1-linked ubiquitin. The study by Tarantino et al. also provided evidence that the NEMO puncta after TNF are associated the receptor complex whereas the NEMO puncta after IL-1 β contain IRAK1 but are not associated with the receptor. Is this also the case in this study? Finally, Tarantino et al. found that NEMO is not required for puncta formation of other components although it is needed for IKK activation. Would this not suggest that phase-separation occurs upstream of NEMO and that the Ub-binding domains of NEMO facilitate the recruitment of NEMO into the structures?

We observed that NEMO-positive puncta in response to both IL-1 β and TNF receptor activation are not limited to the plasma membrane. Based on our recent finding that LUBAC assembles an NF- κ B signaling platform, including NEMO, IKKs and p65, at the mitochondrial outer membrane (Wu et al., EMBO J, in press), we are currently following up the possibility that phase separation of NEMO occurs at mitochondria. In support of this hypothesis,

binding of NEMO to linear polyubiquitin induces and/or selects for a conformational state that promotes membrane association and enhances the affinity for IKK β (Catici et al., 2015).

Specific comments

-It is stated that modification of NEMO by M1-Ub is well-established. However, several studies have failed to detect M1-linked Ub chains on NEMO following immune receptor stimulation, including after IL-1 β treatment. E.g. see Emmerich et al. PNAS 2013, Fiil et al. Mol Cell 2013. The authors should establish if endogenous/exogenous NEMO indeed is M1-Ub-modified after IL-1 β treatment of their cells.

In the revised manuscript, we show that wildtype NEMO is modified by M1-linked ubiquitin upon IL-1 β receptor activation in cells in contrast to the Q330X NEMO mutant (new Fig. 1D). This is also documented in the literature: Emmerich et al. show in their Fig. 1 the assembly of M1-linked polyubiquitin on NEMO upon IL-1 β treatment (Emmerich et al., 2013). Similarly, Tokunaga et al. demonstrated in Fig. 5 of their publication that IL-1 β receptor activation results in M1-linked ubiquitination of NEMO (Tokunaga et al., 2009).

Related to this, it is stated in the discussion that "Ubiquitination of NEMO by HOIP is required for NF- κ B activation in response to various stimuli." I don't think there is experimental evidence that ubiquitination of endogenous NEMO is functionally important for NF- κ B activation. If there are studies demonstrating this, the specific studies and stimuli should be indicated.

There are several studies demonstrating the functional importance of ubiquitination of NEMO for NF- κ B activation, including Smit et al. (Smit et al., 2012), Ikeda et al. (Ikeda et al., 2011); Boisson et al. (Boisson et al., 2012), Niu et al. (Niu et al., 2011), and Tokunaga et al. (Tokunaga et al., 2009). These studies showed that linear ubiquitin chain formation is indispensable for full activation of the canonical NF- κ B pathway and that HOIP conjugates linear polyubiquitin chains to NEMO. Tokunaga et al. reported that the expression of the catalytically inactive HOIP C885S mutant has a dominant negative effect on NF- κ B activation in response to IL-1 β or TNF (Tokunaga et al., 2009). Moreover, IL-1 β -induced NF- κ B activation is significantly reduced in cells expressing the NEMO mutant K275R/K309R, which lacks the acceptor lysines for HOIP-mediated ubiquitination (Tokunaga et al., 2009).

-The use of the Q330X NEMO mutant complicates the interpretation of the data given that the truncation interferes with Ub-binding and NEMO ubiquitination. It would be important to demonstrate that the truncated protein in cells maintains other interaction such interaction with IKKa/IKKb.

In the revised version, we show that Q330X NEMO is not impaired in interacting with IKK β , but does not interact with HOIP and linear ubiquitin chains, although the UBAN domain and the HOIP-interacting region is not affected by the truncation (new Fig. 1D). This behaviour makes the Q330X NEMO mutant even more interesting.

-In Figure 1 and 2 it is suggested that ubiquitination of NEMO by HOIP in vitro promoted phase separation of NEMO. Since NEMO Q330X is not ubiquitinated by HOIP, the most straight forward interpretation is that HOIP ubiquitinates NEMO on residues within the truncated region and not the ub sites (K285, K309) reported in cells. It would be important to establish that the HOIP-induced NEMO ubiquitination in vitro reflects the ubiquitination of NEMO in cells (with regard to NEMO K residues and Ub chain length). Without this, it seems like a "leap of faith" to claim that in vitro ubiquitination of NEMO and promotion of phase-separation reflects the role of NEMO ubiquitination in response to receptor activation in cells. As pointed out above, the lysines for HOIP-mediated ubiquitination of NEMO have been identified by mass spectrometry and their relevance has been confirmed in cells. Cell lines expressing either wildtype NEMO or K275R/K309R NEMO were treated with IL-1 β for 15 min and covalent linkage of NEMO was analysed after lysis under denaturing conditions followed by immunoprecipitation and immunoblotting. Only wildtype but not K275R/K309R NEMO was modified by linear ubiquitin chains under these conditions, see Fig. 5D in Tokunaga et al. (Tokunaga et al., 2009).

References

- Boisson B, Laplantine E, Prando C, Giliiani S, Israelsson E, Xu Z, Abhyankar A, Israel L, Trevejo-Nunez G, Bogunovic D et al. (2012) Immunodeficiency, autoinflammation and amylopectinosis in humans with inherited HOIL-1 and LUBAC deficiency. *Nat Immunol* 13: 1178-86
- Catici DA, Horne JE, Cooper GE, Pudney CR (2015) Polyubiquitin Drives the Molecular Interactions of the NF-kappaB Essential Modulator (NEMO) by Allosteric Regulation. *J Biol Chem* 290: 14130-9
- Du M, Ea CK, Fang Y, Chen ZJ (2022) Liquid phase separation of NEMO induced by polyubiquitin chains activates NF-kappaB. *Mol Cell*
- Emmerich CH, Ordureau A, Strickson S, Arthur JS, Pedrioli PG, Komander D, Cohen P (2013) Activation of the canonical IKK complex by K63/M1-linked hybrid ubiquitin chains. *Proc Natl Acad Sci U S A* 110: 15247-52

Ikeda F, Deribe YL, Skanland SS, Stieglitz B, Grabbe C, Franz-Wachtel M, van Wijk SJ, Goswami P, Nagy V, Terzic J et al. (2011) SHARPIN forms a linear ubiquitin ligase complex regulating NF-kappaB activity and apoptosis. *Nature* 471: 637-41

Ivins FJ, Montgomery MG, Smith SJ, Morris-Davies AC, Taylor IA, Rittinger K (2009) NEMO oligomerization and its ubiquitin-binding properties. *Biochem J* 421: 243-51

Kensche T, Tokunaga F, Ikeda F, Goto E, Iwai K, Dikic I (2012) Analysis of nuclear factor-kappaB (NF-kappaB) essential modulator (NEMO) binding to linear and lysine-linked ubiquitin chains and its role in the activation of NF-kappaB. *J Biol Chem* 287: 23626-34

Komander D, Reyes-Turcu F, Licchesi JD, Odenwaelder P, Wilkinson KD, Barford D (2009) Molecular discrimination of structurally equivalent Lys 63-linked and linear polyubiquitin chains. *EMBO Rep* 10: 466-73

Laplantine E, Fontan E, Chiaravalli J, Lopez T, Lakisic G, Veron M, Agou F, Israel A (2009) NEMO specifically recognizes K63-linked poly-ubiquitin chains through a new bipartite ubiquitin-binding domain. *EMBO J* 28: 2885-95

Lo YC, Lin SC, Rospigliosi CC, Conze DB, Wu CJ, Ashwell JD, Eliezer D, Wu H (2009) Structural basis for recognition of diubiquitins by NEMO. *Mol Cell* 33: 602-15

Niu J, Shi Y, Iwai K, Wu ZH (2011) LUBAC regulates NF-kappaB activation upon genotoxic stress by promoting linear ubiquitination of NEMO. *EMBO J* 30: 3741-53

Rahighi S, Ikeda F, Kawasaki M, Akutsu M, Suzuki N, Kato R, Kensche T, Uejima T, Bloor S, Komander D et al. (2009) Specific recognition of linear ubiquitin chains by NEMO is important for NF-kappaB activation. *Cell* 136: 1098-109

Rahighi S, Iyer M, Oveisi H, Nasser S, Duong V (2022) Structural basis for the simultaneous recognition of NEMO and acceptor ubiquitin by the HOIP NZF1 domain. *Sci Rep* 12: 12241

Smit JJ, Monteferrario D, Noordermeer SM, van Dijk WJ, van der Reijden BA, Sixma TK (2012) The E3 ligase HOIP specifies linear ubiquitin chain assembly through its RING-IBR-RING domain and the unique LDD extension. *EMBO J* 31: 3833-44

Tokunaga F, Sakata S, Saeki Y, Satomi Y, Kirisako T, Kamei K, Nakagawa T, Kato M, Murata S, Yamaoka S et al. (2009) Involvement of linear polyubiquitylation of NEMO in NF-kappaB activation. *Nat Cell Biol* 11: 123-32

Yoshikawa A, Sato Y, Yamashita M, Mimura H, Yamagata A, Fukai S (2009) Crystal structure of the NEMO ubiquitin-binding domain in complex with Lys 63-linked di-ubiquitin. *FEBS Lett* 583: 3317-22

November 21, 2022

Re: Life Science Alliance manuscript #LSA-2022-01607-TR

Prof. Konstanze F Winklhofer
Ruhr University Bochum
Molecular Cell Biology
Universitaetsstrasse 150
Bochum, North Rhine-Westphalia 44801
Germany

Dear Dr. Winklhofer,

Thank you for submitting your revised manuscript entitled "Linear ubiquitination induces NEMO phase separation to activate NF- κ B signaling" to Life Science Alliance. The manuscript has been seen by the original reviewers whose comments are appended below. While the reviewers continue to be overall positive about the work in terms of its suitability for Life Science Alliance, some important issues remain. We, thus, encourage you to submit a revised version of the manuscript back to LSA that specifically address:

- The remaining Reviewer 1's points (in text and figures)
- Reviewer 2's point to prove that LUBAC activity is essential for NEMO clustering/phase separation by running LUBAC inactivation by KO or KD of HOIP in cells and see whether this impairs clustering of NEMO WT upon IL-1 β treatment
- Reviewer 3's points by toning down the here mentioned conclusions and, if feasible, investigate whether endogenous NEMO in the cell systems used is indeed modified by LUBAC / M1-Ub

Our general policy is that papers are considered through only one revision cycle; however, given that the suggested changes are relatively minor, we are open to one additional short round of revision. Please note that I will expect to make a final decision without additional reviewer input upon resubmission.

Please submit the final revision within two months, along with a letter that includes a point by point response to the remaining reviewer comments.

To upload the revised version of your manuscript, please log in to your account: <https://lsa.msubmit.net/cgi-bin/main.plex>
You will be guided to complete the submission of your revised manuscript and to fill in all necessary information.

- A letter addressing the reviewers' comments point by point.
- An editable version of the final text (.DOC or .DOCX) is needed for copyediting (no PDFs).
- High-resolution figure, supplementary figure and video files uploaded as individual files: See our detailed guidelines for preparing your production-ready images, <https://www.life-science-alliance.org/authors>
- Summary blurb (enter in submission system): A short text summarizing in a single sentence the study (max. 200 characters including spaces). This text is used in conjunction with the titles of papers, hence should be informative and complementary to the title and running title. It should describe the context and significance of the findings for a general readership; it should be written in the present tense and refer to the work in the third person. Author names should not be mentioned.

B. MANUSCRIPT ORGANIZATION AND FORMATTING:

Sincerely,

Novella Guidi, PhD

Reviewer #1 (Comments to the Authors (Required)):

This manuscript is much improved over the prior version. The paper discusses some of the molecular determinants involved in mediating phase separation of NEMO with the addition of M1-Ub4 chains, and the authors correlate this behavior with NEMO function inside cells. The authors have addressed many of my previous concerns. Overall, the storyline is much clearer. The work complements/supports recent work on NEMO (Du et al. 2022). Several points need additional clarity:

The authors include Figure 3B to demonstrate the liquid-like behavior. An additional note may be added to the figure legend as these droplet fusion images appear to be taken in solution as opposed to on a coverslip. I mention this because the images in Figure 4, 6, and elsewhere do not show spherical droplets, so I wonder about the material properties. Are these images suggesting coverslip interactions or do these NEMO/M1-Ub droplets mature over time? Change in material properties over time may be the reason why NEMO/M1-Ub4 condensates do not disappear after OTULIN addition as OTULIN may disassemble the soluble M1-Ub4 chains or ubiquitinated NEMO but not the components inside the potentially more solid-like condensates.

A quick comment may be added about whether the phase diagram in Figure 4C is similar to that in Du et al. 2022 Figure 5C.

For Figure 6B, I would be cautious in interpretation of the lack of phase separation for the CoZi-CTD-NEMO and CoZi NEMO domains as it appears these experiments were done at a single set of conditions. To be precise, phase diagrams for these constructs should be collected, although I understand if this is outside the scope of this work.

In Figure 1, Western Blot in panel C has low resolution; will originals be provided in supplementary?

In Figure S1, SEC profiles do appear similar but do both broad peaks in each chromatogram correspond to NEMO? Lanes in the gel in S1B are not labeled, so I'm not sure what these samples correspond to.

On pg. 3 (second paragraph) when 'inter and intramolecular interactions are maximized' is mentioned, I would recommend adding biomolecule-solvent interactions to the list. There is an overemphasis on biomolecule-biomolecule interactions, but solvent is also very important for mediating phase separation.

Reviewer #2 (Comments to the Authors (Required)):

The quality of the manuscript has been strongly improved by adding new data and by restructuring the result section. Even though it does not provide major new insights compared to the recent comprehensive analyses by the Chen lab (PMID: 35477005), it confirms the results and adds some new aspects.

However, in line with my previous general comments, I strongly recommend to add at least one important new aspect to the manuscript. The data strongly suggest that LUBAC activity is essential for NEMO clustering/phase separation. Can the authors show that HOIP KO or KD in cells impairs clustering of NEMO WT upon IL-1b treatment (see Figure 2A). This experiment would provide strong evidence for the in vivo relevance of phase separation.

As an additional point, I think the SEC performed with NEMO WT and Q330X is not allowing to draw a clear conclusion about differences in oligomerization. There is an equal distribution of NEMO WT and Q330X in all fractions, but the WB in Fig. S1B is not labeled properly. It has been shown that recombinant full length NEMO alone is forming higher order oligomers with apparent molecular weight > 400 kDa (PMID 10893415). It seems that a column used here is not sufficient to resolve such oligomers, but that structures of larger 400 kDa need to be separated.

Referee Cross-Comments

I agree with reviewer 3 that it has not been shown that NEMO conjugation by Met1-linked Ub chains is driving NF- κ B signaling. Moreover, recent manuscripts support the idea that the substrate of LUBAC is ubiquitin, meaning that an initiator ubiquitination (mono/poly) on the substrate NEMO needs to take place to allow linear ubiquitin chain assembly. Thus, the involvement and nature of NEMO Ub chains requires a much closer inspection. However, as the authors mention, NEMO is also binding to Met1-linked ubiquitin chains, which can enforce clustering/phase separation. Given the comments of reviewer 3, I believe it will be an important step forward to show that LUBAC inactivation, e.g. by HOIP or HOIL-1 deficiency, abolishes clustering of cellular NEMO. Independent of the mechanism (Ub conjugation versus reversible binding), such data will lend strong support to the claim that clustering/phase separation is important inside the cells.

Reviewer #3 (Comments to the Authors (Required)):

The revised manuscript is improved relative to the original submission to the other journal and the new data included solidify some of the observations of the original manuscript. However, there remains some concerns regarding the manuscript that needs to be addressed. It remains unclear whether or not the NEMO-containing structures in cells reflect phase-separation observed *in vitro* and, critically, whether the structures represent signaling-competent complexes. There is a sense that the authors overinterpret some of the data and/or extrapolate *in vitro* data to cellular signaling mechanisms without providing direct experimental evidence. For example, in the discussion the authors state "Here we show that linear ubiquitination of NEMO and its binding to M1-linked ubiquitin chains functions as a switch to induce phase separation of NEMO, which is a prerequisite for the formation of signaling-competent assemblies." In this reviewer's opinion, the study does not provide evidence for the formation of "signaling-competent assemblies"

A related point is that some conclusions are based too much on selected earlier studies. This is particularly the case regarding the functional importance of M1-Ub modification of NEMO, which is claimed to be required for NF- κ B activation. However, as discussed in detail below, the evidence supporting this is not quite as well-established as the authors claim. In fact, several studies have failed to show M1-Ub modification of endogenous NEMO but rather of other proteins in the receptor signaling pathways activating NF- κ B. Whilst this does not exclude that the authors are correct in their conclusion, this reviewer feels it is essential to demonstrate that endogenous NEMO in the cell systems used is indeed modified by LUBAC / M1-Ub. If only overexpressed NEMO becomes ubiquitinated after IL-1 β treatment, this would put into question the biological relevance of the study.

Specific points relating to the rebuttal:

The authors seem to have misunderstood the experiment shown in Fig. 1 in Emmerich et al. 2013. The study uses recombinant Halo-NEMO to purify K63/M1-hybrid Ub chains. These Ub chains are NOT conjugated to NEMO but rather to other proteins in the IL-1 β pathway (IRAK1, IRAK4, MYD88, etc). In fact, in the results section the authors state "...indicating that NEMO is not modified significantly with M1-Ub after IL-1 stimulation."

The authors correctly point out that Tokunaga et al. 2009 and Niu et al. 2011 provide evidence that M1-Ub of ectopically expressed NEMO contributes to NF- κ B activation after IL-1 β and Etoposide treatment, respectively. However, the other cited studies do not address the functional importance of M1-Ub conjugated to NEMO. Since the study by Tokunaga et al., several receptor associated proteins have been proposed to be important targets for M1-Ub chain formation, including RIP1 and TNFR1 after TNF, IRAK1/IRAK4/MYD88 after IL-1 β , and RIP2 after NOD2 (Gerlach et al. 2011, Draper et al. 2015, Emmerich et al. 2013, 2016, Damgaard et al. 2012, Fiil et al. 2013). Several of these studies have failed to detect M1-Ub-modified NEMO although other proteins were clearly modified by M1-Ub (e.g. see Emmerich et al. 2013, Emmerich 2016, Damgaard et al. 2012). This does NOT exclude that NEMO is a substrate for LUBAC but there is quite a lot of evidence indicating that NEMO might not be the primary substrate for LUBAC and there is in this reviewer's opinion not sufficient experimental data to support the statement: "Ubiquitination of NEMO by HOIP is required for NF- κ B activation in response to various stimuli."

The authors refer to K275/K309 (should be K285/K309) as Ub-sites on NEMO that were identified in Tokunaga et al. 2009. K285 and K309 have indeed been found to be modified after TNF in Wagner et al. 2016 but it is also worth noting that Ikeda et al. 2011 show that LUBAC activity mediates (directly or indirectly) ubiquitination of several other sites on NEMO, many of which are lysine residues downstream of position 330 (see Ikeda et al. 2011, Suppl. Figure 5). Hence, it would be important for the mechanism proposed to determine which sites LUBAC modifies *in vitro* and IF these are the same NEMO Ub sites being modified in the cellular assays. The positioning of the M1-Ub on NEMO may well affect assembly of condensates / phase-separation. The authors could consider introducing the K285R, K309R mutations and assess if these mutations prevent the ubiquitination by LUBAC and phase separation. This might provide functional evidence that NEMO ubiquitination contributes to phase separation *in vitro* and in cells.

As reviewer #2 suggests, assessing phase-separation using HOIP KO or KD cells could provide functional insights into NEMO condensates although it should be noted that M1-Ub contributes to the overall assembly of receptor signalling complexes (Haas et al. 2009).

Following the suggestion of reviewers 2 and 3, we included HOIP CRISPR/Cas9 knockout (KO) SH-SY5Y cells in our study, which we recently generated in our lab. In HOIP-deficient SH-SY5Y cells, IL-1 β -induced cytoplasmic foci formation of endogenous NEMO was not observed, in contrast to control SH-SY5Y cells (new Fig. 2C, left panel). A corresponding immunoblot indicated that IKK complex activation upon IL-1 β receptor activation is impaired in the absence of HOIP (new Fig. 2C, right panel). These experiments suggested that the IL-1 β -induced formation of NEMO assemblies and efficient downstream activation of NF- κ B requires HOIP-dependent formation of linear ubiquitin chains.

In addition, we tested the K285/K309R NEMO mutant, as suggested by reviewer 3. This mutant lacks two lysine residues for covalent linear ubiquitin chain attachment (Tokunaga, 2009). We expressed the K285/K309R NEMO mutant in an NEMO-deficient background and observed that in response to IL-1 β receptor activation only wildtype NEMO but neither K285/K309R NEMO nor Q330X NEMO formed foci (new Fig. 2A, left panel). Loss of NEMO foci formation was accompanied by defective IKK complex activation (new Fig. 2A, left panel).

Point-by-Point response

Reviewer #1

This manuscript is much improved over the prior version. The paper discusses some of the molecular determinants involved in mediating phase separation of NEMO with the addition of M1-Ub4 chains, and the authors correlate this behavior with NEMO function inside cells. The authors have addressed many of my previous concerns. Overall, the storyline is much clearer. The work complements/supports recent work on NEMO (Du et al. 2022).

Several points need additional clarity:

The authors include Figure 3B to demonstrate the liquid-like behavior. An additional note may be added to the figure legend as these droplet fusion images appear to be taken in solution as opposed to on a coverslip. I mention this because the images in Figure 4, 6, and elsewhere do not show spherical droplets, so I wonder about the material properties. Are these images suggesting coverslip interactions or do these NEMO/M1-Ub droplets mature over time? Change in material properties over time may be the reason why NEMO/M1-Ub4 condensates do not disappear after OTULIN addition as OTULIN may disassemble the

soluble M1-Ub4 chains or ubiquitinated NEMO but not the components inside the potentially more solid-like condensates.

The droplet fusion images shown in Fig. 3B were also taken on coverslips, similarly to the images shown in Fig. 4 and 6. The difference in appearance is probably due to the magnification. The scale bar has been added to Fig. 3B for clarification.

Indeed, we were also considering the possibility that OTULIN cannot reach the droplets due to changes in material properties, but the blue gel shown in Fig. 4F indicated that OTULIN completely degraded the 4xM1-ubiquitin chains (lanes 4 and 5). Thus, OTULIN was obviously able to enter the droplets and hydrolyze M1-linked ubiquitin inside the droplets. However, we agree that changes in material properties over time could play a role for the persistence of NEMO condensates and included this aspect in our discussion.

A quick comment may be added about whether the phase diagram in Figure 4C is similar to that in Du et al. 2022 Figure 5C.

The phase diagram shown in Fig. 4C is similar to that shown in Fig. 5C of Du et al. 2022, for which K63-linked polyubiquitin was used. We mentioned this in the revised manuscript.

For Figure 6B, I would be cautious in interpretation of the lack of phase separation for the CoZi-CTD-NEMO and CoZi NEMO domains as it appears these experiments were done at a single set of conditions. To be precise, phase diagrams for these constructs should be collected, although I understand if this is outside the scope of this work.

We fully agree, but what we can conclude is that CoZi-CTD NEMO and CoZi NEMO does not undergo phase separation at a concentration of 5 μ M, where wildtype NEMO does. We also tested other conditions. At higher concentrations, CoZi-CTD NEMO did not phase separate, whereas CoZi NEMO started to aggregate beyond a concentration of 50 μ M.

In Figure 1, Western Blot in panel C has low resolution; will originals be provided in supplementary?

We have replaced the blot by a higher resolution blot. Of course, the original blots will be provided as source data.

In Figure S1, SEC profiles do appear similar but do both broad peaks in each chromatogram correspond to NEMO? Lanes in the gel in S1B are not labeled, so I'm not sure what these samples correspond to.

We are sorry for the confusion. Both broad peaks correspond to NEMO as size exclusion chromatography (SEC) was performed after affinity purification of the expressed proteins. In

the revised version, the gels corresponding to each peak are shown and the labeling of the Coomassie gels have been modified for clarification and to correspond with the eluted fractions displayed on the profile peaks.

On pg. 3 (second paragraph) when 'inter and intramolecular interactions are maximized' is mentioned, I would recommend adding biomolecule-solvent interactions to the list. There is an overemphasis on biomolecule-biomolecule interactions, but solvent is also very important for mediating phase separation.

Yes, we fully agree. The important aspect of biomolecule-solvent interactions has been included in the revised version.

Reviewer #2

The quality of the manuscript has been strongly improved by adding new data and by restructuring the result section. Even though it does not provide major new insights compared to the recent comprehensive analyses by the Chen lab (PMID: 35477005), it confirms the results and adds some new aspects.

However, in line with my previous general comments, I strongly recommend to add at least one important new aspect to the manuscript. The data strongly suggest that LUBAC activity is essential for NEMO clustering/phase separation. Can the authors show that HOIP KO or KD in cells impairs clustering of NEMO WT upon IL-1 β treatment (see Figure 2A). This experiment would provide strong evidence for the in vivo relevance of phase separation. We have recently generated HOIP KO SH-SY5Y cells by CRISPR/Cas9 technology, which we used for the experiment suggested by reviewer 2 and 3. Upon IL-1 β treatment, endogenous NEMO formed foci in control SH-SY5Y cells, whereas no foci formation was observed in HOIP-deficient cells (new Fig. 2C, left panel). The lack of NEMO foci formation was accompanied by defective IKK complex activation (new Fig. 2C, right panel). These results confirmed the correlation between NEMO foci formation and NF- κ B pathway activation upon IL-1 β receptor activation, supporting the relevance of NEMO phase separation in cells.

As an additional point, I think the SEC performed with NEMO WT and Q330X is not allowing to draw a clear conclusion about differences in oligomerization. There is an equal distribution of NEMO WT and Q330X in all fractions, but the WB in Fig. S1B is not labeled properly. It has been shown that recombinant full length NEMO alone is forming higher order oligomers with apparent molecular weight > 400 kDa (PMID 10893415). It seems that a column used

here is not sufficient to resolve such oligomers, but that structures of larger 400 kDa need to be separated.

Size exclusion chromatography (SEC) was performed after affinity purification of the expressed proteins. The SEC profiles show an early peak, which is barely in the calibrated range of the SEC column used. Therefore, we cannot resolve oligomeric/aggregated NEMO species, as noted by this reviewer. However, the intention of this SEC was to compare wildtype and Q330X NEMO and it turned out that both show a similar profile. The NEMO monomer is visible later in the chromatogram and is labelled accordingly in the revised version.

Reviewer #3

The revised manuscript is improved relative to the original submission to the other journal and the new data included solidify some of the observations of the original manuscript. However, there remains some concerns regarding the manuscript that needs to be addressed. It remains unclear whether or not the NEMO-containing structures in cells reflect phase-separation observed in vitro and, critically, whether the structures represent signaling-competent complexes. There is a sense that the authors overinterpret some of the data and/or extrapolate in vitro data to cellular signaling mechanisms without providing direct experimental evidence. For example, in the discussion the authors state "Here we show that linear ubiquitination of NEMO and its binding to M1-linked ubiquitin chains functions as a switch to induce phase separation of NEMO, which is a prerequisite for the formation of signaling-competent assemblies." In this reviewer's opinion, the study does not provide evidence for the formation of "signaling-competent assemblies"

In the revised version, we added more data, indicating that NEMO foci formation and IKK complex activation are at least correlated in cell. The analysis of HOIP CRISPR/Cas9 KO cells and the K285/309R NEMO mutant confirmed that defective NEMO foci formation is accompanied by impaired IKK complex activation (new Fig. 2A and 2C).

A related point is that some conclusions are based too much on selected earlier studies. This is particularly the case regarding the functional importance of M1-Ub modification of NEMO, which is claimed to be required for NF- κ B activation. However, as discussed in detail below, the evidence supporting this is not quite as well-established as the authors claim. In fact, several studies have failed to show M1-Ub modification of endogenous NEMO but rather of other proteins in the receptor signaling pathways activating NF- κ B. Whilst this does not exclude that the authors are correct in their conclusion, this reviewer feels it is essential to demonstrate that endogenous NEMO in the cell systems used is indeed modified by

LUBAC / M1-Ub. If only overexpressed NEMO becomes ubiquitinated after IL-1 β treatment, this would put into question the biological relevance of the study.

We appreciate the reviewer's comment on this issue and therefore rephrased all paragraphs on covalent ubiquitination of NEMO by LUBAC. It was not our intention to insist on a crucial role of NEMO M1-ubiquitination in IL-1 β -induced NF- κ B pathway activation. The reason why we stressed this initially, was to bring up a new aspect on M1-ubiquitin-induced phase separation, which was not addressed by Du et al. (2022), who studied the effect of NEMO binding to K63- and M1-linked ubiquitin chains. Nevertheless, we observed that the K285/309R NEMO mutant does neither form foci nor promote IKK complex activation upon IL-1 β treatment (new Fig. 2A).

Specific points relating to the rebuttal:

The authors seem to have misunderstood the experiment shown in Fig. 1 in Emmerich et al. 2013. The study uses recombinant Halo-NEMO to purify K63/M1-hybrid Ub chains. These Ub chains are NOT conjugated to NEMO but rather to other proteins in the IL-1 β pathway (IRAK1, IRAK4, MYD88, etc). In fact, in the results section the authors state "...indicating that NEMO is not modified significantly with M1-Ub after IL-1 stimulation."

The authors correctly point out that Tokunaga et al. 2009 and Niu et al. 2011 provide evidence that M1-Ub of ectopically expressed NEMO contributes to NF- κ B activation after IL-1 β and Etoposide treatment, respectively. However, the other cited studies do not address the functional importance of M1-Ub conjugated to NEMO. Since the study by Tokunaga et al., several receptors associated proteins have been proposed to be important targets for M1-Ub chain formation, including RIP1 and TNFR1 after TNF, IRAK1/IRAK4/MYD88 after IL-1 β , and RIP2 after NOD2 (Gerlach et al. 2011, Draper et al. 2015, Emmerich et al. 2013, 2016, Damgaard et al. 2012, Fiil et al. 2013). Several of these studies have failed to detect M1-Ub-modified NEMO although other proteins were clearly modified by M1-Ub (e.g. see Emmerich et al. 2013, Emmerich 2016, Damgaard et al. 2012). This does NOT exclude that NEMO is a substrate for LUBAC but there is quite a lot of evidence indicating that NEMO might not be the primary substrate for LUBAC and there is in this reviewer's opinion not sufficient experimental data to support the statement:

"Ubiquitination of NEMO by HOIP is required for NF- κ B activation in response to various stimuli."

As already mentioned above, we agree and therefore we modified the revised version accordingly.

The authors refer to K275/K309 (should be K285/K309) as Ub-sites on NEMO that were identified in Tokunaga et al. 2009. K285 and K309 have indeed been found to be modified

after TNF in Wagner et al. 2016 but it is also worth noting that Ikeda et al. 2011 show that LUBAC activity mediates (directly or indirectly) ubiquitination of several other sites on NEMO, many of which are lysine residues downstream of position 330 (see Ikeda et al. 2011, Suppl. Figure 5). Hence, it would be important for the mechanism proposed to determine which sites LUBAC modifies in vitro and IF these are the same NEMO Ub sites being modified in the cellular assays. The positioning of the M1-Ub on NEMO may well affect assembly of condensates / phase-separation. The authors could consider introducing the K285R, K309R mutations and assess if these mutations prevent the ubiquitination by LUBAC and phase separation. This might provide functional evidence that NEMO ubiquitination contributes to phase separation in vitro and in cells.

We thank reviewer 3 for this suggestion. In the revised version, we included data on the K285/309R NEMO mutant, which we expressed in an NEMO-deficient background. Similarly to the Q330X NEMO mutant, the K285/309R NEMO mutant did not form foci in response to IL-1 β receptor stimulation and did not promote IKK complex activation (new Fig. 2A).

As reviewer #2 suggests, assessing phase-separation using HOIP KO or KD cells could provide functional insights into NEMO condensates although it should be noted that M1-Ub contributes to the overall assembly of receptor signalling complexes (Haas et al. 2009).

As outlined above, we included this experiment in the revised version (new Fig. 2C).

January 3, 2023

RE: Life Science Alliance Manuscript #LSA-2022-01607-TRR

Prof. Konstanze F Winklhofer
Ruhr University Bochum
Molecular Cell Biology
Universitaetsstrasse 150
Bochum, North Rhine-Westphalia 44801
Germany

Dear Dr. Winklhofer,

Thank you for submitting your revised manuscript entitled "Linear ubiquitination induces NEMO phase separation to activate NF- κ B signaling". We would be happy to publish your paper in Life Science Alliance pending final revisions necessary to meet our formatting guidelines.

- please make sure that the author order in the manuscript and our system match and that each author is entered in our system
- please add a figure callout for figure S2E to your main manuscript text
- please provide original blots as Source Data, with one Source Data Figure per main figure

A. FINAL FILES:

B. MANUSCRIPT ORGANIZATION AND FORMATTING:

Sincerely,

January 9, 2023

RE: Life Science Alliance Manuscript #LSA-2022-01607-TRRR

Prof. Konstanze F Winklhofer
Ruhr University Bochum
Molecular Cell Biology
Universitaetsstrasse 150
Bochum, North Rhine-Westphalia 44801
Germany

Dear Dr. Winklhofer,

Thank you for submitting your Research Article entitled "Linear ubiquitination induces NEMO phase separation to activate NF- κ B signaling". It is a pleasure to let you know that your manuscript is now accepted for publication in Life Science Alliance. Congratulations on this interesting work.

DISTRIBUTION OF MATERIALS:

Again, congratulations on a very nice paper. I hope you found the review process to be constructive and are pleased with how the manuscript was handled editorially. We look forward to future exciting submissions from your lab.

Sincerely,
